# Unveiling the Dynamics behind Glioblastoma Multiforme Single-Cell Data Heterogeneity

**DOI:** 10.3390/ijms25094894

**Published:** 2024-04-30

**Authors:** Marcos Guilherme Vieira Junior , Adriano Maurício de Almeida Côrtes, Flávia Raquel Gonçalves Carneiro, Nicolas Carels, Fabrício Alves Barbosa da Silva

**Affiliations:** 1Graduate Program in Computational and Systems Biology, Oswaldo Cruz Institute (IOC), Oswaldo Cruz Foundation (FIOCRUZ), Rio de Janeiro 21040-900, Brazil; marcosvieira.research@gmail.com; 2Department of Applied Mathematics, Institute of Mathematics, Federal University of Rio de Janeiro (UFRJ), Rio de Janeiro 21941-909, Brazil; adricortes@cos.ufrj.br; 3Systems Engineering and Computer Science Program, Coordination of Postgraduate Programs in Engineering (COPPE), Federal University of Rio de Janeiro (UFRJ), Rio de Janeiro 21941-972, Brazil; 4Center of Technological Development in Health (CDTS), Oswaldo Cruz Foundation (FIOCRUZ), Rio de Janeiro 21040-361, Brazil; flavia.carneiro@fiocruz.br; 5Laboratório Interdisciplinar de Pesquisas Médicas, Oswaldo Cruz Institute (IOC), Oswaldo Cruz Foundation (FIOCRUZ), Rio de Janeiro 21040-900, Brazil; 6Program of Immunology and Tumor Biology, Brazilian National Cancer Institute (INCA), Rio de Janeiro 20231-050, Brazil; 7Laboratory of Biological System Modeling, Center of Technological Development in Health (CDTS), Oswaldo Cruz Foundation (FIOCRUZ), Rio de Janeiro 21040-361, Brazil; 8Scientific Computing Program, Oswaldo Cruz Foundation (FIOCRUZ), Rio de Janeiro 21041-222, Brazil

**Keywords:** Glioblastoma Multiforme, epigenetic landscape, parameter sets estimation, single-cell RNA sequencing, heterogeneity, cancer attractors, gene regulatory network dynamics

## Abstract

Glioblastoma Multiforme is a brain tumor distinguished by its aggressiveness. We suggested that this aggressiveness leads single-cell RNA-sequence data (scRNA-seq) to span a representative portion of the cancer attractors domain. This conjecture allowed us to interpret the scRNA-seq heterogeneity as reflecting a representative trajectory within the attractor’s domain. We considered factors such as genomic instability to characterize the cancer dynamics through stochastic fixed points. The fixed points were derived from centroids obtained through various clustering methods to verify our method sensitivity. This methodological foundation is based upon sample and time average equivalence, assigning an interpretative value to the data cluster centroids and supporting parameters estimation. We used stochastic simulations to reproduce the dynamics, and our results showed an alignment between experimental and simulated dataset centroids. We also computed the Waddington landscape, which provided a visual framework for validating the centroids and standard deviations as characterizations of cancer attractors. Additionally, we examined the stability and transitions between attractors and revealed a potential interplay between subtypes. These transitions might be related to cancer recurrence and progression, connecting the molecular mechanisms of cancer heterogeneity with statistical properties of gene expression dynamics. Our work advances the modeling of gene expression dynamics and paves the way for personalized therapeutic interventions.

## 1. Introduction

Notwithstanding the significant advancements in understanding and therapeutics, cancer continues to be a predominant global cause of mortality [1]. Glioblastoma multiforme (GBM; Appendix C) stands as the most common and aggressive brain tumor, characterized by an average survival time of 15 months and a roughly 10% probability of achieving a 5-year overall survival [2,3]. Single-cell RNA sequencing (scRNA-seq) has spotlighted the pronounced heterogeneity inherent in GBM [4,5]. Intriguingly, this level of complexity in scRNA-seq data is not exclusive to GBM. Many cancers manifest similar intricate patterns in their sequencing data [6,7], underpinning a broader challenge in oncology research. With such heterogeneity possibly driving the aggressiveness of these malignancies [8,9], pressing questions emerge regarding the underlying dynamics. Foremost among these are the interpretations of observed data distributions, their ensuing consequences, and the patterns encapsulated within the data.

Extensive research has delved into the complexities of carcinogenesis, advancing discourses on driver and passenger mutations and the profound influences of epigenetics [10,11,12]. In the intricate landscape of gene regulatory networks (GRN) dynamics, pivotal studies have elucidated the alignment between cell types or subtypes and stable states, often termed ‘attractors’ [13,14]. Concurrently, certain oscillatory cellular processes—integral to diverse functions such as circadian rhythms [15], cell cycle progression [16], and NF-κB signaling in response to inflammation [17]—are closely tied to limit cycles in GRNs. Yet, the foundational principles guiding transitions between these states and the mechanisms by which a system traverses within an attractor’s domain—a notion sometimes framed as a ‘cancer attractor’ [18]—remain areas of active research.

The intricacies of tumor evolutionary trajectories further underscore a pressing need for understanding. As cancer progresses or counteracts therapeutic measures, the dynamic shifts in tumor subclonal architectures come to the forefront [19]. Traditional linear evolution models may inadequately capture the complexities of tumor evolution. A dominant clone proliferates in certain scenarios, producing a predominantly homogeneous tumor composition. Conversely, other situations present coexisting subclonal populations, suggesting a more branched evolutionary pattern than a purely linear trajectory [20]. In light of these challenges, the increasing availability of omics data presents an opportunity for deeper investigations. For instance, studies have identified subtypes of GBM, a fundamental leap forward in understanding the disease and a critical step for choosing the applying treatment [2,21,22,23]. Yet, as the horizon of our knowledge expands, questions about the evolutionary pathways of these subtypes and the dynamic interplay underpinning their classifications persist, beckoning deeper investigations [24,25,26].

This evolving scenario necessitates a shift in perspective. Instead of remaining anchored in reductionist perspectives, there is a pressing call to embrace a more systemic approach. This perspective provides a comprehensive view of cell type and functionality dynamics and is reinforced by studies such as [27]. This systemic thinking is present in contemporary paradigms that depict cancer as a nuanced series of events leading to a ’state disease’ [28], rather than being merely the fallout of isolated mutations. In this context, we define the *state* as the biochemical milieu of a cell, signifying the evolutionary trajectory of states within a complex system. Contrasting starkly with earlier models that portrayed cancer progression as linear, this perspective revels in understanding the multifaceted dynamics of the disease within a broader, multidimensional context [13]. A vivid analogy for this conceptual shift can be traced back to C. Waddington’s 1957 metaphor, where cellular differentiation is visualized as a sphere traversing an (epi)genetic landscape (since the RNA sequencing data provide an estimate of expression after genetic and epigenetic regulations, and it is not possible to verify the contributions of each explicitly, we chose to use parentheses in (epi)genetics) of peaks and valleys [29], with the zeniths representing the undifferentiated phenotype.

In the Waddington (epi)genetic landscape, a differentiated state is achieved through developmental paths influenced by both intrinsic cellular factors—as the cell’s historical events, such as lineage, gene expression patterns, and epigenetic modifications [30]—and by extrinsic factors such as tissue-level perturbations or environmental influences. According to this perspective, cancer might be associated with one or more (defective or malignant) attractor states in the (epi)genetic landscape. These malignant attractors can either pre-exist hidden within the landscape or emerge due to genetic and epigenetic alterations. In both scenarios, the attractors are undesirably reached due to the inherent stochastic fluctuations of biological systems at the biomolecular scale [18].

Using the Waddington (epi)genetic landscape as a conceptual framework allows us to understand the extent of heterogeneity within GBM more clearly. As Waddington envisioned, cellular differentiation is deeply intertwined with the nuances of cancer progression. If we extend this framework, it is clear that the various paths and trajectories through which a cell might journey—and eventually culminate in a malignant phenotype—are likely shaped by a combination of genetic, epigenetic, and environmental forces. This understanding reinforces the need to decipher the intricate details behind single-cell data, especially since heterogeneity is both an outcome and an influencer of tumor dynamics. As we delve deeper, the landscape metaphor becomes more than just a conceptual tool; it provides a practical framework that guides our investigation of GBM. By examining how and why certain trajectories or states become prominent in GBM, we lay the groundwork for exploring the underlying factors and mechanisms that drive cellular heterogeneity and what this might signify for our broader understanding of cancer evolution.

From this exploration, it becomes evident that cellular heterogeneity in single-cell data raises crucial questions. These questions concern the interplay between epigenetic regulation, genomic stability, selective pressures, and the inherent GRN governing cancer cell behavior. A thorough investigation into these aspects offers fresh perspectives into GBM’s evolving landscape. One of the primary sources of heterogeneity is genomic instability [20]. This instability, marked by a high mutation rate at the DNA level, results in a cellular milieu teeming with diversity [31]. While genomic instability significantly contributes to heterogeneity, it represents only one facet of the complexity. Genetic alterations and epigenetic mechanisms lead to diverse cellular responses, amplifying the heterogeneity and adding layers to our understanding. This way, the dynamics underlying heterogeneity are molded by additional factors adding complexity.

Selective pressures exert a significant influence on cancer heterogeneity. Rather than being merely passive, these pressures actively shape the outcome of the diversity introduced by genomic instability [32]. They curate the cellular environment, favoring the emergence of stable cellular states and modulating the dynamics within the (epi)genetic landscape. Such stable states, called ‘attractors’, represent characteristic cellular phenotypes. Trajectories within the phase space tend to gravitate towards these attractors, shaped by the so-called ’basins of attraction’. In light of this, the interplay between the chaos of genomic instability and the order introduced by selective pressures, as revealed through the lens of clonal evolution, offers a detailed understanding [32]. It is not merely about heterogeneity; it underscores a structured heterogeneity, reflecting a complex dynamic infused with order.

This narrative implies a representation that captures a cell’s transformative journey toward a malignant state. This work proposes modeling this representation that aligns with the snapshot-like data from scRNA-seq (Figure 1). Stemming from a structured stochastic limit cycle [33] (Figure 1I), the cellular trajectory is subject to perturbations from genetic mutations, epigenetic regulations, and selective pressures. We highlight that some dimensions, potentially associated with marker genes, may already exhibit a fixed-point dynamic even in this initial state. As these forces act, the initial trajectory undergoes alterations, manifested as increased stochastic noise that broadens the oscillatory boundary, or ‘stochastic tube’ (Figure 1II). This amplification in noise effectively populates the state space, thereby increasing the relevance of mean values as accurate representations of the cellular dynamics. Finally, the cell adopts more erratic behavior, eventually taking on characteristics resembling a random wandering around a fixed point (Figure 1III). This transformation might occur for multiple genes, culminating in a tumor’s genetic and phenotypic variability, termed intratumor heterogeneity. Importantly, this evolving subclonal architecture is dynamic, undergoing continual shifts throughout disease progression, and thus, presenting challenges for both diagnostics and therapeutic strategies.

Models aiming to represent a cell’s transformative journey, especially through the lens of the (epi)genetic landscape, have emerged in the scientific community [14,34,35,36,37,38,39,40]. These models provide insights into cellular states’ temporal flow and stability, enhancing our understanding of cancer’s evolutionary pathways. However, they face challenges owing to their reliance on time series data for parameter estimation—a requirement often elusive given the intricacies of capturing real-time biological processes. Recognizing these limitations, techniques have emerged that infer temporality (the ordered succession of events) through pseudotime [41]. Though pseudotime provides a promising avenue to infer transitions between attractors using scRNA-seq data, it is not without its challenges. Events such as mutations, deregulation of the cell cycle [42], and cellular heterogeneity can potentially obscure the interpretation of pseudotime trajectories, further complicating the characterization of cancer dynamics. Yet, such complexities underscore the relevance of a stochastic modeling approach.

Temporal data acquisition presents inherent challenges, giving rise to numerous theoretical hypotheses. Among these, a central theme is a system’s statistical behavior, tracking its evolution in phase space over time. A distinguishing feature of some of these perspectives is the convergence of time averages to ensemble averages [43], which is particularly important for snapshot data such as scRNA-seq. This approach offers an avenue to bypass the intricacies of temporal sampling. Expanding on this, some theories emphasize examining specific components within a system, such as basins of attraction [44,45]. Translating this to the GBM context, the distinct aggressiveness levels of each subtype can be interpreted as different moments in a broader cellular narrative. We propose that the ensemble averages derived from GBM subtypes might be akin to time averages of a representative trajectory (Figure 1). This correlation might be especially evident in specific gene space dimensions, likely marker genes. Combined with the fixed points modeling, such intertwining of theoretical insights with observed biological phenomena has inspired our hypothesis, driving us toward a comprehensive modeling perspective.

Given these challenges and insights, our hypothesis is grounded in the idea of a system that, over an extended timeframe, will navigate through all accessible states within the bounds of a cancer attractor with a consistent likelihood. While this assumption might simplify certain complexities, it establishes a robust foundation to probe the stochastic essence of single-cell data. Such a model gathers the concepts of an equilibrium state with inherent variability, offers a streamlined approach to estimate parameters, and interprets the variability presented in data. However, should there be complications in estimating these parameters, we may need to reassess the foundation of our hypothesis. Validity would largely depend on comparing simulated outputs with experimental findings, a step that could significantly enhance our hypothesis’s reliability.

In validating our hypothesis, we employ experimental GBM scRNA-seq data to generate a computational model, incorporating principles from (epi)genetic landscape modeling, Langevin’s dynamics [46], and Hill Functions [47] with modifications for the GRN dynamics. Our methodology centers on dynamic modeling, integrating raw data with insights derived from the underlying biological processes and mechanisms. In the proposed context, the cancer attractor concept suggests a propensity for specific cancer subtypes, recognizable as distinctive regions in phase space. Additionally, while exploring the gene expression space, some areas are expected to be inaccessible due to biological constraints. Our assumptions include metric transitivity, which means that two points in phase space can be connected by a shortest path in the gene expression space. This phenomenon aligns with the idea that intrinsic cellular noise enhances phase space exploration in cancer cells, diminishing the barriers between different basins of attraction [18]. Building upon these insights into the gene expression space and its constraints, we propose a stochastic model in silico, aiming to quantify the (epi)genetic landscapes derived from scRNA-seq data of four GBM subtypes: Classical, Mesenchymal, Proneural, and Neural. This model also contemplates interactions inherent to a GBM-specific GRN.

Our modeling efforts promise more than theoretical research. For instance, they hint at tangible avenues for interpreting the intricacies of genomic instability related to cancer heterogeneity. As various mechanisms that contribute to genomic instability imprint distinct molecular signatures [20,31], by exploring the statistical behaviors explained in our approach, we could potentially correlate mechanisms underlying these signatures, offering a chance to identify novel therapeutic targets [48]. In other words, we aimed to associate alterations in statistical properties observed in scRNA-seq data with molecular-level events that modify the system’s exploration of possible states. Distinguishing these unique molecular imprints would allow us to forge stronger connections between the ‘geometry of heterogeneity’ seen in single-cell data and distinct instability mechanisms, offering a more enriched understanding of tumor biology. Consequently, this statistical property could be viewed as a consequence of the progression of the malignant state.

In light of these insights and their challenges, this report seeks to integrate the theoretical foundations of the Waddington (epi)genetic landscape with the wealth of data emerging from single-cell technologies. By leveraging a stochastic dynamics model, we aimed to unravel the intricate mechanisms that sculpt the heterogeneity inherent to the GBM landscape. Our methodology provides a data-driven quantification of the (epi)genetic landscape specific to GBM and its respective subtypes. Additionally, we probed the statistical dynamics of our in silico model, establishing a framework for subsequent inquiries and potential practical applications. Furthermore, it contributes to developing studies on a biological system’s possible long-term behavior and stability.

## 2. Results

The results section is organized as follows: (i) We present the gene regulatory network, which defines the vertices (genes/TFs) and edges (regulation interactions). (ii) We discuss the initial challenges of choosing a clustering method, which will subsequently influence parameter estimation and optimization. One of the outputs of this optimization process is the selection of regulation functions, which we present in the following. (iii) We compare experimental data with various simulated data obtained using different clustering methods. This comparison serves not only to assess the accuracy of the parameter estimates, but also to provide a valuable perspective on the simulation outcomes. (iv) We examine a chosen case from the simulations to verify the dynamics inside the basins of attraction and the hypothesis of equivalence between the sample and time average. The results may also serve as a preliminary study for analyzing experimental data as they become more available. Finally, we aim to comprehensively understand the relationships between data analysis, parameter estimation, regulation functions, and simulations by presenting the results in this order.

### 2.1. Glioblastoma GRN

Our initial step was constructing a GRN, with the methodology outlined in Appendix A. This network, visualized in Figure 2, captures the intricate interplay among key genes and markers related to GBM subtypes. Primarily based on an initial list of GBM markers, the MetaCore platform autonomously expanded the network, ensuring an objective, bias-free augmentation. The resulting structure comprises 40 vertices and 242 edges: 187 activations, 11 self-activations, 41 inhibitions, and 3 self-inhibitions. Appendix A details the vertices, including those initially selected and those added by MetaCore.

### 2.2. Clustering Methods and Parameters Estimation

A central question in the scRNA-seq analysis is how to interpret gene expression variations across individual cells. Initial data analysis revealed a group of genes with apparently multimodal distributions (see Appendix A). Such patterns, inherent in complex systems such as GBM, possibly hint at various cellular states, emphasizing the tumor’s heterogeneous nature. This observation led us to investigate the extent to which this pattern represented the presence of multiple clusters (see Appendix A). Focusing on the four pivotal marker genes delineating GBM subtypes (as described in Section 4.4), our analytical approach employed dimensionality reduction using t-SNE in Wolfram Mathematica. With a perplexity of 60, this method facilitated an optimal visualization of potential clusters, as depicted in Appendix A.

We utilized two different clustering methods (k-means and NbC) to perform the clustering analysis. We configured the built-in Mathematica functions for both clustering methods with the ‘PerformanceGoal’ set to quality, the ‘CriterionFunction’ set to standard deviation, and the ‘DistanceFunction’ set to Euclidean distance. By comparing the results of these two clustering methods, we aimed to understand the underlying data structure and identify the optimal number of clusters for the given gene expression data. All clusters’ statistics are available in the Appendix A.

Parameter estimation was conducted using the centroid coordinates of all genes within each cluster in an oversized fit, considering k=1. All auxiliary parameter values are available in the Appendix A. An interval ranging from 0.01 to 10 was used to adjust the parameters. These values were initially established through manual checks, ensuring that changes in the interval would not lead to substantial variations in the quality estimator values (differences in the interval resulting in approximately 1% changes in the indicator values were disregarded). The first estimate was performed using Equation (Equation 18), obtaining 784 sets of parameters. The distribution of values for each parameter and both clustering methods can be seen in Appendix A. As shown in the figure, some parameters have distributions that vary within the interval limits, such as a[*EGR1*] and a[*SOX4*]. In contrast, others display values constrained to smaller intervals, such as a[*E2F1*] and a[*TP53*]. This pattern is evident in both clustering methods, signifying a strong relationship with the network structure. The larger variations in specific parameters can demonstrate how these parameters are susceptible to changes within the established network. In contrast, others may require considerable compensation, without which it could lead to undesirable changes. The second parameter estimation using Equation (Equation 19) was performed on top of the first to adjust the values for each interaction. This step’s distribution can be seen in Appendix A. The distribution of most parameters is around the unitary value, suggesting a possible correction of the previous estimate. This observation implies that the second estimation step fine-tunes the parameters to enhance the model’s accuracy.

Appendix A show the residual for each gene for the two clustering methods superimposed, one for the first parameters estimation and the other for the second parameter estimation. Both figures demonstrate that the two consecutive parameter estimations did not significantly affect the residuals of the first parameter estimation and indicate that it did not result in overfitting. Additionally, the model showed good compatibility for some genes based on the centroids of the experimental data clusters. For instance, genes such as *HIF1A* and *SOX2* may have presented low residuals due to their close centroid values. In contrast, genes such as *CD44* and *EGFR* exhibited high residuals and variations due to their distant centroid values. While these residuals may indicate a need for adjustments in network interactions or the model itself, they might also reflect the influence of the clustering method. Therefore, analyzing the relationship between residuals, network structure, and the clustering method is crucial for drawing more accurate conclusions. Furthermore, this analysis could provide insights into the model’s limitations and help identify potential improvement areas. We present all residual values and statistics in the Appendix A.

The final step of the parameter estimation process aimed to assess the compatibility between the cluster centroids derived from experimental data and those obtained from parameter adjustments. To accomplish this, we selected the parameters with the smallest *T* and *R* values (as displayed in Table 1). Appendix A illustrate the activation (left) and inhibition (right) matrices, with the color gradient signifying the logarithm of the parameter values for both clustering methods. The ‘o’ and ‘x’ symbols denote the regulations and self-regulations present in the network, respectively. Parameter values are acquired by multiplying parameters from the first and second fits (columns and rows weighting). The simulation proceeds by multiplying each matrix element by either 0 or 1 (1 if the element corresponds to an ‘x’ or ‘o’ position, and 0 otherwise) to account for only the combinations present in the network. Appendix A displays the ceiling function (ceil()) of all parameter values for k-means clustering and NbC clustering, respectively. The ceiling function, ceil(), rounds up a given number to the nearest integer. In this case, it helps visualize the order of magnitude of the parameter values. The left images correspond to activation parameters, while the right images depict inhibition parameters. All parameter values are available in the Appendix A.

The global optimization process aimed to maximize the number of simulated gene distributions compatible with the experimental data by optimizing the global strength of activation, self-activation, inhibition, and self-inhibition (‘x’ and ‘o’ positions) using multiplicative factors. The factors were chosen to maximize the average number of genes that stay within two sigmas of their cluster centroids. In the first and second parameter estimation steps, 784 sets were generated for both k-means and Neighborhood Contraction clustering methods. The best sets, number 435 for k-means clustering and number 428 for Neighborhood Contraction clustering were then used in the optimization process. Appendix A illustrate the distribution of genes compatible with each attractor during the stochastic parameter analysis for both clustering methods.

For the k-means clustering, the optimization process yielded a list of 191 values, with the best set being number 150 (out of 191). The experimental attractors were defined as A, B, C, D, and E. After varying the regulation weights in each stochastic simulation, we got attractors A and C with an average of about 15 genes within two sigmas in relation to the observed data, attractor B with an average greater than 20, attractor D with an average close to 20, and attractor E with an average of only 5 genes compatible with the data. Moreover, the following compatibility with data values was observed: attractor A (15), B (24), C (30), D (29), and E (4) when using factors 1.4, 0.7, 0.9, and 1.3, respectively. In the case of Neighborhood Contraction clustering, the optimization process generated a list of 13 values. The best set was number 11 (out of 13), and the clustering ranged from A to G. The compatibility with data values for attractors A to G were 19, 20, 11, 20, 17, 17, and 13 when using factors 1.4, 0.6, 1.3, and 1.2, respectively. Table 1 presents the values of two parameter sets: one for k-means clustering (5 clusters) and another for Neighborhood Contraction (7 clusters).

When comparing the k-means and Neighborhood Contraction clustering methods, we observed differences in the number of genes compatible with each attractor and the quality of the estimated parameters. For k-means clustering, the total number of parameters that lead to genes compatible with attractors A, B, C, D, and E was 191, while 13 with attractors A to G for Neighborhood Contraction clustering. These results suggest that as the number of clusters increases, the set of parameters that matches the data for the established conditions decreases.

### 2.3. Regulation Functions

One of the results of the first and second estimations was the selection of the regulation function. The modified regulation function depends on the transcription factor and the regulated genes. The total number of regulatory functions equals the number of interactions in the network (242 interactions). After the parameters optimization, the best compatibility between experimental and simulated data was obtained with the regulation function of Equation (Equation 15). For example, Appendix A shows the activation and inhibition regulation functions for some gene/transcription factor combinations. Each case presents a different relationship between the variables representing the amount of transcription factor mRNA and the activity of the gene promoter. Appendix A correspond to the regulation functions of the first clustering choice, while Appendix A to the second choice. Figure 3 displays some of these cases, such as the activation of *CHI3L1* by *SP1* and the inhibition of *EGFR* by *STAT3*. All surfaces were obtained with n=1 and $h_{2}\left(x\right)$, with (a) and (b) using $f_{a}=0.1$ and $f_{b}=1.3$ and (c) and (d) using $f_{a}=0.1$ and $f_{b}=1.1$.

The dependence of the regulation function on the transcription factor can be seen along its axis, as affected by the contribution of the parameter *f* ($f_{a}$ for activation and $f_{b}$ for inhibition) and the values $\left(\max_{\alpha }\;\left[X_{j}^{\alpha }\right]+1\right)$. The dependence on the target gene is represented in the transverse direction to the transcription factor axis, where the maximum values for constant transcription factor concentration correspond to the centroid values observed in the experimental data. It is possible to observe how the peaks occur around the average values of *CD44*, *CHI3L1*, *EGFR*, and *IDH1* for activation functions and *CD44*, *EGFR*, *MDM2*, and *PGK1* for inhibition (Appendix A).

Considering constant target gene conditions, the regulation functions recover one-dimensional curves, similar to standard Hill functions with n=1. At high target levels, these functions approach zero, mirroring trends in our experimental data. This pattern mimics potential biological mechanisms not explicitly detailed in the network, such as missing environmental signals or missing interactions. Notably, lower inactivation values mean greater inhibitions; a smaller $V^{b}$ value indicates increased inhibition and decreased basal activation. Furthermore, the modified regulation function does not exhibit peaks at zero target concentrations, a deliberate change to prevent unwanted activation peaks that might affect observed null data values. Beyond these specific cases, the system’s progression will be guided by the interplay of network interactions, parameter values, and noise.

### 2.4. Comparing Experimental and Simulated Data

#### 2.4.1. Noise and Distributions Compatibility

Data transformation often results in a dependence of the standard deviation on the mean [49]. Part of this dependency can be reduced depending on the normalization process [50]. However, to achieve a better fit between the simulation outcomes and the sequencing data, the function for the multiplicative noise was defined as Equation (Equation 26), with *p* = 0.23 and $c_{1}$ = 8, which were found empirically to fit the data best. Appendix A presents the graphic for these parameter values. The optimized set of parameters was evaluated by comparing the histograms of expression levels for data at different time points to assess whether the system had reached a steady state. Appendix A compares experimental and simulated data. Appendix A shows the simulation outcome for time 50 (500 steps), Appendix A shows time 25 (250 steps), Appendix A shows time 5 (50 steps), and Appendix A shows the initial conditions. The means of the simulated distributions are mostly within two standard deviations of the means of the observed multimodal distributions in the experimental data, demonstrating good compatibility for most genes. However, mainly for higher expression values, the simulated outcomes exhibit smaller standard deviations than those observed experimentally. Gene expression values for each of the three times are available in the Appendix A.

#### 2.4.2. Clusters Compatibility

To evaluate the congruence between patterns in simulated and experimental data, we employed k-means [51] and Neighborhood Contraction (NbC) [52] on the experimental data and extended the analysis to include Gaussian Mixture [53] for the simulated datasets. The clustering outcomes are depicted in Appendix A, which represents data simulated from parameters estimated after the k-means and NbC clustering of experimental samples. Clusters were obtained in two reduced dimensions instead of four original marker gene dimensions, facilitating a more tractable visual assessment. Preliminary analyses suggest NbC’s superior performance in the reduced dimensional space for both simulation cases.

To advance our comparison, we assessed centroids of simulated data juxtaposed with data distributions. Appendix A illustrate these for k-means and NbC methods. The centroids, regardless of method, particularly between NbC and Gaussian Mixture, displayed notable similarities. These centroids from the simulated data appeared to represent the centers of the experimental data’s multimodal distributions more precisely than direct clustering. Such findings could underscore biologically pertinent insights if these distributions reflect core biological activities.

To further elucidate our findings, we analyzed the proportion of data points in each cluster using pie charts from both experimental and simulated data. Appendix A illustrate these distributions. Clusters A and B demonstrate consistent proportions across most scenarios, except when using the Gaussian Mixture on NbC simulated data. Other clusters displayed method-specific variations, possibly influenced by metastable clusters. Such clusters can challenge clustering techniques, leading to misclassification or indistinct cluster boundaries. Since clustering methods may address metastable clusters differently, discrepancies arise in both qualitative and quantitative outcomes. Consequently, the chosen clustering method is critical in determining clustering results.

We also assessed the distribution within each cluster. Appendix A indicated that standard deviations in every simulated data cluster were consistently lower than in experimental data, suggesting narrow local stability influenced by estimated parameters, and potential noise-induced jumps between attractors. Through boxplots and histograms, we contrasted gene expression distributions within clusters. As evidenced by Appendix A, the most congruent gene expression distributions were observed in clusters B, C, and E for post-k-means, and A, B, and G for post-NbC. Appendix A present the specific distributions, excluding *NEFL* due to its near-zero distribution. These findings emphasize our earlier observations about centroid alignment and standard deviation differences. Parameter estimates using NbC produced divergent simulated clustering outcomes, indicating result instability.

In analyzing scatter plots of the combined marker genes, excluding the *NEFL* gene, transitional points between clusters were evident. As depicted in Appendix A, these points might sustain the narrow stability but potentially challenge the sample-time average equivalence if they arise frequently. As these points denote distinct cells at the same time step, further exploration of single trajectories over time is essential. Another observation is that, despite the differences in distribution dispersion, a qualitative analysis (low and high) of the clusters would have compatible results. This suggests that the clusters may still be comparable when focusing on their overall trends rather than the specific dispersion of data points.

#### 2.4.3. Experimental and Simulated Landscapes

Considering the results obtained, we now examine the corresponding (epi)genetic landscapes. Assuming that each cluster’s average and deviation characterize each cell state’s distribution, visualizing a potential surface derived directly from the experimental data can help evaluate the clustering quality and serve as a reference for a qualitative comparison with the simulation results. Appendix A present the landscapes for experimental and simulated data clustering methods. A constant of 0.01 was added due to the presence of null values (or close to zero) for the standard deviation of some genes. This value was sufficient to avoid numerical issues and was significantly smaller than nonnull values.

Qualitative changes can be observed due to the greater number of attractors in the NbC method compared to the k-means one, particularly in the dimensions *CD44* × *IDH1* (Figure 4a,b), *CD44* × *NEFL*, and *EGFR* × *IDH1*, with the presence of new basins. When comparing these experimental landscapes to the simulated ones (Figure 4c,d), as expected from our previous discussion of Section 2.4.2, there are significant differences. The landscape is a precise structure that requires a high degree of correspondence between means, standard deviations, weights, and attractors. It is important to note that the experimental landscape does not consider dynamics and takes into account data points that may be in transient or metastable states. Regardless, the previous results are relevant and may indicate avenues for further improvements and studies on the system dynamics.

### 2.5. Dynamics Inside Basins for the Chosen Simulated Case

The results and analysis presented above lay the groundwork for further investigations into various topics related to the system’s dynamics and properties. Here, we will explore results concerning the statistical dynamics inside the basins of attraction. First, let us examine a collection of trajectories originating from the centroids of the clustered simulated data. Then, we selected one case to investigate the dynamic properties of the trajectories. This case corresponds to clustering using a Gaussian mixture after the parameters’ estimation of the k-means clustered data. Appendix A displays the trajectories starting from each cluster centroid for each gene, using the parameters obtained after k-means clustering and the Gaussian mixture cluster classification.

For most clusters and genes, the trajectories oscillate around their respective centroids. However, for these single trajectories of each cluster, we can observe that specific genes, such as *CD44* and *EGFR*, exhibit transitions (Figure 5). Under the sample and time average equivalence hypothesis, such transitions may not be frequent, occurring on a longer timescale than the component dynamics. This suggests that a more detailed investigation is needed to better understand the nature and frequency of these transitions in the context of the system’s statistical properties.

To investigate the transitions between components further, we focused on two main features: identifying and quantifying the transitions among the defined basins of attraction and the time spent in each basin before the occurrence of a transition. This analysis was performed by sampling the time steps of each trajectory starting from the centroid of each cluster, measuring each point’s Euclidean distances to all basins, and assigning it to the closest one. We also considered establishing a threshold to determine whether a point belongs to a basin. Still, it would lead to challenges associated with complex dynamic systems and basins of attraction. In addition, determining the basins’ exact boundaries and neighborhood sizes can be difficult, as the basins’ shapes may be irregular and not easily described by simple measures such as distances or standard deviations. Ultimately, we decided to focus on the closest basin assignment, acknowledging the limitations and complexities involved in this approach.

We sampled 100 trajectories for each cluster and tracked their paths. Appendix A show the quantification of the transitions between the basins. Absent basins indicate that there were no transitions to or from them. Appendix A displays the number of transitions between each cluster, which may be affected by multiple transitions within a single trajectory. The most frequent transitions were observed between clusters A and C, followed by D to A. Both of these transitions are visible in Appendix A. Appendix A quantifies the probability of a jump from one cluster to another. The frequencies were normalized by the total outgoing events. For instance, considering the system at cluster A, the figure indicates a 0.85 probability of jumping to cluster C and 0.15 to cluster D. We chose this normalization to allow for the prediction of which cluster the system is more likely to transition to given the present state. An alternative would be pairwise in and out normalization, e.g., A goes more frequently to C than C comes to A. However, given the system is in a specific state, it would not inform the expected transition.

Studying the time spent by a trajectory within each cluster can help us understand the system’s stability and the relative significance of each identified state. Long residence times within a cluster suggest a stable state, while shorter times could indicate a transient or metastable state. Appendix A displays the time spent in each cluster before it jumps to another. The absence of dispersion in clusters C and E suggests that the trajectories starting in these clusters remained there. This observation does not contradict the transition graph; it simply means that when jumping to cluster C and returning to their original cluster, we might consider that the transition was not fully completed, as the system may not have reached the narrow stability of cluster C. This could be due to significant fluctuations in other dimensions. In any case, we can observe that the time spent before transitioning is relatively high for all basins compared to the total observation time.

Additionally, Appendix A shows the histogram for the number of transitions within each trajectory. In other words, considering the 500 trajectories (100 for each cluster), most did not exhibit any transition. The most frequent number of transitions was 1, followed by a decrease in frequency as the number of transitions increased. These results agree with the hypothesis of low-frequency transitions. However, quantifying a timescale within each basin and basin transitions is still required to verify the hypothesis and further understand the system’s behavior.

Before assessing the timescale, let us consider Appendix A, illustrating the dispersion of trajectory time points for each basin. We can see the well-defined localization and even the time points possibly representing transitions between clusters. The points are all time points for three trajectories of each cluster. Figure 6 summarizes the results, with (a) one of the 3D trajectories, (b) the transition matrix, (c) the frequency of transitions per trajectory, and (d) the time spent before a transition. The 3D plots help visualize the complexity of each trajectory, and it is important to remember that they are a simplification of the 40-dimensional space. This analysis provides a qualitative intuition, given that the biological system is much more complex.

Studying autocorrelations within the trajectories is essential to better understanding the system’s dynamics and gaining insights into the timescales. Autocorrelation analysis can reveal the system’s temporal structure. By examining these structures, we can verify timescales that characterize each basin’s internal dynamics or are associated with transitions between basins. Furthermore, this approach allows us to connect the qualitative intuition provided by the trajectory plots with quantitative measures that can more accurately describe the system’s statistical behavior.

Appendix A shows the time series side by side with its corresponding autocorrelation functions for all genes and basins considering two sampled trajectories. The autocorrelation functions represent the autocorrelation for each time lag up to a specific maximum lag. The autocorrelation function can be a powerful tool for understanding the characteristic timescales of the system’s dynamics. By analyzing the autocorrelation function, we can identify the timescales at which the system exhibits significant correlations, indicating the persistence of certain behaviors or states. In addition, we can see that each basin and gene may present different timescales, with consistency for both simulations. Besides the visual inspection, we defined another way to quantify this due to the data complexity. We computed the timescale as the minimum time step to reach an autocorrelation value below $e^{-1}$ for each basin, variable, and simulation.

Appendix A aim to understand the distribution of these timescales within each internal dynamic and search for possible transition behavior. Appendix A shows that even with timescales varying between genes, they tend to present similar values for each gene. However, compared to other clusters’ timescale patterns, some discrepant cases are observed for *CD44* and *EGFR* genes and for cluster E. We can see in Appendix A that these cases presented transitions, leading to increased timescales. Figure 7 displays an example of the increased timescale for the *CD44* by comparing the autocorrelation with and without the presence of a transition. These figures highlight the different characteristic timescales within the basins and inter-basins.

Furthermore, Appendix A emphasize the values in each basin. Appendix A shows the significant variations of timescales for the genes within each basin. Still, given the previous discussion, we note that some genes present very narrow timescale distributions while others have wide error bars. This suggests that the latter may be related to transitions and could be potential variables for further analysis.

Finally, Appendix A shows that the average over all genes and trajectories yields very close results, with only cluster E displaying a different pattern. This may suggest that the timescales are interconnected, and the system may somehow compensate for them. The underlying biological processes within the system may lead to interconnectedness that allows for compensation across different timescales. This observation opens up opportunities for further investigation into the mechanisms behind this behavior and how it might relate to the system’s overall function and stability.

After conducting all these analyses, we can finally assess the compatibility between time and sample averages. Appendix A demonstrates this compatibility for nearly all clusters and genes. The left panels of Appendix A represent the average of 100 samples at the final time. In contrast, the right panels display the time average, considering 10 trajectories from time 30 to 50 (steps 300 to 500). Once again, the observed discrepancies, such as in *CD44* basin B and D, might be connected to transitions between clusters (Figure 8). We computed these averages considering their departure states, and they may have jumped to another basin during the process.

In conclusion, these results support our initial assumption of the statistical dynamics behavior inside cancer attractor basins, even if only for simulations. The correspondence with the data suggests that it might also be true for biological systems. This finding encourages further investigation of these properties within biological samples. Our analyses can be compared and utilized for future applications, ultimately contributing to a better understanding of the system and developing more effective treatments.

## 3. Discussion

In an effort to examine scRNA-seq datasets through stochastic modeling, we encountered challenges ranging from a lack of time series data to complex network construction and a vast number of parameters. Here, we discuss our results and interpretation in the context of the present and possible future contributions.

### 3.1. Dynamics of GBM: Aggressiveness and Heterogeneity

In this report, we introduced an approach to investigate the dynamics underlying tumor heterogeneity as revealed by scRNA-seq data of GBM cancer cells. We strategically focused on GBM due to its pronounced aggressiveness, which allowed us to make specific assumptions about inherent dynamics. Recognizing GBM’s aggressive nature, we assumed our sample contained a considerable portion of the cancer attractor domain. We proposed that factors such as genomic instability, epigenetic changes, and selective pressures could alter inherent trajectories, yielding diverse dynamic behaviors. For instance, stochastic fixed points might become increasingly noisy, or a stochastic limit cycle could either resemble a stochastic fixed point or bifurcate towards it (Figure 1).

To our knowledge, this report represents the first application of an integrated framework combining clustering techniques to identify stochastic fixed points and model the cancer attractors dynamics of GBM subtypes using scRNA-seq data. This methodology provided conceptual support for the empirical application of the cancer attractor model. In essence, while our dataset provides a snapshot, this framework provides an underlying theory that shapes this snapshot. This scenario provided a context so we could consider the cluster centroid (sample mean) equivalent to the time average of a representative trajectory within the cancer attractor domain—each cluster symbolizing a distinct cancer subtype phenotype.

We also delineated the clusters and identified the centroids by examining a reduced dimension of specific marker genes, as highlighted in studies such as Sáez et al. [54]. We proposed that these reduced dimensions are particularly amenable to the manifestation of stochastic fixed point dynamics, owing to the inherent characteristics that render them suitable for subtype classification. Our analyses indicated that expression values (both experimental and simulated) are indeed constrained within subregions of these marker gene dimensions, which proved instrumental in studying the dynamics of subtypes. These results reinforced the importance of these subtypes biomarkers. Despite the inability to capture the full breadth of experimental data distributions, our model demonstrates that scRNA-seq data can, to a significant degree, be characterized by our theoretical construct, a premise we plan to expand upon in subsequent sections. We regard this as a progressive step in improving the validity of our hypothesis and demonstrating the viability of our modeling approach to elucidate the dynamics underpinning the experimental data.

### 3.2. GBM GRN Dynamics: Refining the Model and Centroid-Based Parameter Tuning

Our modeling approach aimed to recreate observed data with stochastic simulations to test how our hypothesis could account for the data patterns. We initiated our model by establishing a GRN and setting up its dynamics. For our GRN, we focused on marker genes and employed the MetaCore platform to establish their interconnections. Although this GRN is a simplified model of the complete biological system, we hypothesized that the dynamics of these marker genes would sufficiently encapsulate the subtypes’ dynamics. We proposed modifications to the Hill functions to better reflect these dynamics. Integrating average values from gene expression clusters into the regulatory model was a key enhancement. This strategy helped to enclose a range of conditions not explicitly included in the model, such as epigenetic differences in patient samples, genetic mutations, missing GRN interactions, or influences from the tumor environment. Our method was designed to construct a dynamics model that closely aligns with experimental data while attenuating the network structure’s sensitivity. By adapting the Hill functions, we overcame inherent limitations and successfully captured the centroids observed among different patients. This method proved to be an efficient and effective way to simplify the network’s complexity by incorporating these variations through the data.

Building upon the improved Hill functions, linear programming was employed to find the parameters that interpolate the dynamics passing through the centroids in biomarker dimensions. As shown in Appendix A, patients’ data follow similar distribution patterns. Based on this observation, we proposed that a more accurate characterization of the disease landscape could be achieved by considering all patient data to improve the sampling around each cancer attractor. Additionally, tuning the model parameters to fit the centroids of all clusters simultaneously was instrumental in preventing the model from being overfitted to any single cluster. It also allowed us to explore whether a single set of parameters could effectively capture intersecting dynamics between all patients and subtypes. In other words, if different subtypes could coexist within the same biological landscape. Indeed, our findings demonstrated that we successfully identified a set of parameters that not only captured the centroids of the data but also delineated a possible dynamic of the disease subtypes. This aligns with findings by [4], which demonstrate the plasticity of GBM cells and their potential for a single cell to give rise to multiple subtypes. These insights suggest a disease landscape where such cellular states are not only present but also dynamic, with the potential for transitions between subtypes that may contribute to the progression of the disease under certain conditions. Additionally, the selected parameter values leading to these conditions may point to specific, potentially latent configurational states of epigenetic regulation that permit such transitions. In this sense, identifying a singular parameter set operating as an effective representation of biological phenomena signifies underlying uniformity within the biological dynamics. This attribute could exist beyond inter-patient differences. Lastly, using a simplified GRN in our hypothesis does not limit these findings. More complex networks would likely present even more parameter possibilities, of which our model represents a mere subset.

Building upon these deterministic parameters, stochastic features were introduced to match the centroids and distributions observed in the experimental dataset. The challenge lies in the precise calibration of this noise. We sought a noise that neither dominates the dynamics nor is negligible. By achieving this, we assert that the noise model, albeit simplistic, serves as an effective representation. Recognizing the complex factors contributing to this noise, ranging from intrinsic cellular mechanisms to varied patient-specific influences, an effective noise encapsulation arises as a pragmatic choice. Furthermore, the study’s aim was not to trace individual cellular trajectories but to discern broader statistical patterns that emerge from the collective trajectories. Since we are trying to observe the ’subtypes envelope’, a prior noise was used to get the first idea of the results before its improvement. This iterative process allows for a progressive refinement of the noise model in future investigations.

### 3.3. Parameters Variability: Heterogeneity and Genomic Instability

As discussed, we have expanded upon traditional cell-type clustering by incorporating dynamic profiling, highlighting the transient nature of cellular states. Our results accentuated the importance of biomarkers, which proved instrumental in delineating stable states and enabled a more subtle understanding of subtype dynamics. However, while our model successfully replicated the centroids of these classifications, it did not achieve the same accuracy for the dispersions. This section focuses on the challenges faced due to the variability of parameters and the interpretation of the underlying causes, including heterogeneity and genomic instability. Furthermore, their impact on modeling heterogeneity in cancer, including genomic instability. These features are intimately associated with cancer and are directly reflected in scRNA-seq data. They can be incorporated into the gene expression variability, both inter- and intra-clusters.

These mechanisms introduce variability that makes dynamical modeling difficult, messing with the parameters and hyperparameters, making its unique determination difficult. As pointed out in [55,56], multiple parameter combinations could yield results closely mirroring those observed experimentally. We adopted statistical criteria to analyze and select the most meaningful parameters to address this multiplicity. Additionally, by changing the hyperparameters, we constructed an extensive parameter analysis that gave us insights into the distribution of these parameters. In this way, the heterogeneity both complicates and helps parameter estimation. It complicates in the sense that the complete variability probably does not reflect a single parameter set but helps in a way that might end up defining the region of the gene expression space that could contain the centroid of the cancer attractor.

By using different hyperparameters to find the best regulation parameters, we obtained parameter distributions for each gene. These distributions may be more than due to methodological change; they could be about intrinsic biological characteristics associated with the genes and their GRN topology. Genes showing wide variability in parameter values across different hyperparameter settings might indicate that the behavior of these genes is highly sensitive to changes in their regulatory environment. Multiple sets of parameters that lead to a driven force near zero reflect the nature of heterogeneity, showing that different cells would display different but near expression values but still could lie around the same attractor. Genes with parameter values that remain relatively consistent across different hyperparameters might be considered more robust or stable in their behavior. This robustness might be due to GRN’s built-in compensatory mechanisms. In the genomic context, it suggests that these gene regulatory networks have evolved to maintain their function despite external perturbations. When such a gene does mutate in cancer, the mutation might have profound effects, given that the gene’s behavior is typically so consistent. In these cases, deviations from this narrow window can destabilize the system, potentially leading the cell into aberrant behaviors. This fact might be intimately related to genomic instability.

Additionally, such genes and the GRN regions they comprise might be less resilient to perturbations and more susceptible to disruptions. This highlights where certain genes can act as points of vulnerability within the network, predisposing the system to disequilibrium and chaotic behaviors when altered. This could be an important feature of this modeling approach, revealing potential genes and regulatory mechanisms to study, new diagnostic pathways, and targeted therapies. However, this fact still needs further verification, increasing sampling size and variability, using a validated network, and comparing the results with biological experiments.

Our parameter modeling offers a robust enhancement to biological interpretation, addressing the limitations of prior studies that often rely on arbitrary parameter values [35,36,57]. Furthermore, we demonstrated that residuals, traditionally utilized as indicators for estimation quality, hold potential as network and model goodness measures. This proposition needs comprehensive exploration in subsequent research efforts. Notably, the presence of nonzero residuals was anticipated, asserting that an exact equilibrium characterization would necessitate the incorporation of multiple complex elements, rendering a precise depiction implausible.

### 3.4. Landscape and Dynamics inside Basins

Central to our investigation is the complexity of modeling dispersions, the varied number of basins, and the emergence of distinct phenotypes. An in-depth comparison of experimental and simulated data enabled us to estimate our model’s performance and predictive capability. The observed deviation from the expected dispersion of experimental data could underscore the need to explore multiple parameter sets following the stochastic fixed-point dynamics. Moreover, technical noise in experimental data—unrelated to core biological processes—might have masked genuine dispersion that our simulated landscape struggled to reproduce. However, an important feature is that the experimental and simulated landscapes identified multiple attractors, reinforcing the hypothesis of multiple stable states. Another point is that the GRN and parameter values might change due to mutations and epigenetic modifications. In that case, it would still support the investigation of short-term dynamics characterization and introduce a possible avenue for investigating tumor progression through longitudinal analysis.

In the final stage of the investigation, we chose suitable parameters. We investigated the in silico dynamics inside each basin, aiming to gain new insights by induction from simulated data. This analysis involved examining the stability of each cluster, comparing time and sample averages, and evaluating the time spent within each cluster, which can serve as a measure of stability. By investigating transitions between attractors, we aimed to understand their stability and the interplay between sample averages and time averages, especially since frequent transitions could challenge the equivalence of these two metrics. To ascertain this, we assessed the frequency of transitions and the timescales within each attractor to better understand the system’s dynamics. It also helped to verify the extent to which the stochastic fixed point could achieve sufficient stability to accommodate the wide experimental data distributions.

In conclusion, we stress the value of studying these aspects in more detail by identifying the factors influencing the system’s properties. If cancer aggressiveness causes a shift leading to a statistical dynamics regime as hypothesized, investigating the detailed causality behind this could provide critical insights into cancer progression and potential interventions. To our knowledge, the in silico verification of stochastic fixed point dynamics and transitions between GBM attractors using this integrated approach has not yet been reported.

## 4. Materials and Methods

Historically, biology has utilized models to interpret complex biological phenomena. Traditional approaches often employed model organisms or cell lines for in vitro studies. However, with recent advancements in computational methods and mathematics, there has been a notable shift towards abstract mathematical models. These models, acting as approximations, allow researchers to navigate and hypothesize within controlled digital environments, simulating the complexities of biological systems. While in silico experiments may not conclusively validate general biological principles, they offer insights into hypothesis outcomes and afford preliminary validation via induction [58].

Gene regulation is one of the most intricate and fundamental biological processes that benefit from computational modeling. Given its inherent complexity and dynamic nature, mathematical modeling has emerged as an indispensable tool for elucidating its nuances.

### 4.1. Model Background

The regulation of gene expression is a complex process that involves multiple layers and mechanisms [13]. One possible measurement of gene expression is the number of messenger RNA (mRNA) molecules that effectively translate into proteins. The expression profile is a dynamic feature, changing in time according to cell types and characteristics. By considering a vector $X=(X_{1},X_{2},\ldots ,X_{N})$, with *N* being the total number of variables and each vector component representing the quantification of mRNA molecules, the cell state can be modeled using system dynamics theory. The basis of the gene regulation dynamics modeling is its associated deterministic differential equations system, an autonomous system of ordinary differential equations $\dot{X}=F\left(X\right)$, containing information about the temporal trajectory driven by the interaction forces between each of its components [59].

There are several possible functions to parameterize the interactions of a nonlinear model, but the common choice is sigmoidal functions. Among them, the Hill function is the most frequent as it has many experimentally observed required characteristics [47]. An example of a driving force *F* using Hill functions can be seen in Wang et al. [34], with a more general form described by:(1)$$F_{i}=-k_{i}X_{i}+\sum_{j\in A_{i}}\frac{a_{ij}X_{j}^{n_{ij}}}{S_{ij}^{n_{ij}}+X_{j}^{n_{ij}}}+\sum_{j\in I_{i}}\frac{b_{ij}{S_{ij}}^{n_{ij}}}{S_{ij}^{n_{ij}}+X_{j}^{n_{ij}}},$$
where, for each gene *i*, represented by the component $X_{i}$, the index sets $A_{i}$ and $I_{i}$ represent the genes that interact with gene *i* through activation and inhibition, respectively. The value *j* represents the edge that bridges the regulation of transcription factors interacting with their target gene promoters. Note that in the case of self-activation or self-inhibition, one has $i\in A_{i}$ or $i\in I_{i}$, respectively. The parameter *S* denotes the value where the Hill function reaches its maximum inclination, *n* represents the intensity of the transition, *a* is the activation coefficient, *b* is the inhibition coefficient, and *k* is the self-degradation constant. When *a* is a self-activation, or *b* is a self-inhibition parameter, they will be denoted by sa and sb, respectively. The parameters *k*, *a*, and *b* have units of ${\mathrm{time}}^{-1}$, while the remaining parameters are dimensionless.

As seen in Equation (Equation 1), the gene activation (*a*) and inhibition (*b*) parameters may vary for each interaction or even as a function of time (nonautonomous system). In addition, sigmoid coefficients may (i) be constant, (ii) vary according to interactions or some proposed functions, or (iii) present a time dependence. In this model, the gene inhibition is given by constraining its basal expression, as can be seen in the positive sign of the inhibition term, with the higher inhibitions obtained by lower values of *b*.

Although using deterministic differential equations to study general behavior is adequate, biological systems are inherently stochastic [59]. Thermal fluctuations and varying conditions affect the likelihood of interactions and make these systems probabilistic. Consequently, the number of molecules over time follows a fluctuating, noisy pattern. A common way to model this stochasticity is through the Chemical Master Equation (CME), a Markovian model that captures the probabilistic nature of molecular interactions [46]. However, solving the CME can be computationally challenging, especially for large systems. An alternative is to use Langevin dynamics [46], which serves as an approximation of the CME, described by a deterministic term F(x) and a stochastic term ξ(t). In Langevin dynamics, we can treat ξ(t) as random fluctuations (without memory) due to its much smaller timescale compared to F(x). The dynamics became:(2)$$\frac{dx(t)}{dt}=F\left(x\right)+\xi \left(t\right)\;,$$
where x(t) is the expression level as a function of time (implicit dependence) relative to random variables of X, F(x) is the deterministic term representing regulation due to network interactions, and ξ(t) is the stochastic term with average 〈ξ(t)〉=0 and amplitude given by its autocorrelation function $\langle \xi_{i}\left(t\right)\xi_{j}\left(t^{\prime }\right)\rangle =2D_{ij}\delta_{ij}\delta \left(t-t^{\prime }\right)$ [37], with *D* being the diffusion coefficient and representing a fluctuation scale factor.

With the presence of fluctuations, probability distributions model gene expression levels, and the temporal evolution p(x,t) is described by the Fokker–Planck Equation (Equation (Equation 35)). This equation provides a continuous approximation to the CME [59], capturing molecular diffusion kinetics across an epigenetic landscape. However, since the Fokker–Planck equation’s driving force and noise components are unknown, an alternative approach to studying the system is to consider a stochastic differential Equation (SDE). In this context, we consider a system described by [14]:(3)dX=μ(X,t)dt+σ(X,t)dW,
where μ(X,t) is the drift, σ(X,t) is the noise parameter, and dW is the Wiener standard process. We assumed μ(X,t)=F(X) by considering the drift due to the driven force. The noise σ(X,t) can be divided into two major contributions: (i) intrinsic, which is related to the system’s internal dynamics, and (ii) extrinsic, which is due to the effects of the environment/microenvironment. We considered a multiplicative noise σ(X,t)=g(X,t) so that the fluctuations may be described by different timescales and constrained by the defined regulation function. The complete definition of the system is given by the parameters of the deterministic and stochastic components, with the specifics of the multiplicative noise and the regulation function being detailed later in this report.

### 4.2. scRNA-Seq Data

While theoretical models provide a conceptual procedure to understand gene regulation, capturing accurate data remains paramount. In recent years, scRNA-seq has emerged as a powerful tool to study gene expression profiles at the individual cell level, enabling the investigation of cellular heterogeneity and the identification of distinct cell subpopulations. This technology has been particularly valuable for studying GRNs. It provides insights into the complex interactions between genes and the possible regulatory mechanisms that drive cell-type-specific gene expression patterns.

#### 4.2.1. GBM and Single-Cell Data

Transitioning from the broader picture of scRNA-seq to its specialized utility, GBM stands out as a compelling case study. GBM, renowned for its profound cellular heterogeneity, exemplifies the challenges researchers grapple with when studying complex disease landscapes [5]. Considering the nuances of cellular evolution in tumor environments, a deeper dive into the roles of selective pressures and genomic instability in shaping GBM’s intricate heterogeneity became imperative. Yet, it is precisely this complexity that makes GBM a fertile ground for scRNA-seq explorations. In this context, single-cell datasets serve as “temporal snapshots”, chronicling the multifaceted expression patterns of GBM’s cellular ensemble at distinct timeline intervals. Although these snapshots might appear isolated, a deeper dive reveals they often resonate with the broader dynamism governing cellular behavior. Figure 1 encapsulates this idea of the richness of information each “snapshot” brings to the table. It suggests that while each scRNA-seq dataset offers a temporally distinct perspective, collectively, they can traverse the entire phase space, capturing the essence of GBM’s intricate dynamics over time. Such insights emphasize scRNA-seq’s transformative potential in unveiling tumor heterogeneity dynamics.

#### 4.2.2. GBM Dataset

We utilized the dataset curated and analyzed by Darmanis et al. [60], which encompasses single-cell resolution RNA sequencing outputs from patients diagnosed with diverse GBM subtypes. The study scrutinized tumor heterogeneity, contrasting the tumor core with its periphery. This dataset aggregates samples from four patients, all diagnosed with primary GBM and characterized by a negative *IDH1* signature (indicating an absence of mutations in the *IDH* gene). Following stringent quality control measures, the dataset retained information from 3589 cells, including various cell types from the central nervous system, such as vascular, immune, neuronal, and glial cells.

The analytical framework employed by Darmanis et al. [60] identified cellular clusters from the dimensionality reduction with tSNE, layered over a dissimilarity matrix. Subsequent clustering via the k-means algorithm refined cellular groupings. A meticulous gene expression audit identified the signature genes of each cluster, the results of which were cross-referenced against healthy tissue data to chart cellular identities. Residual clusters were cataloged as neoplastic, predominantly localized to the tumor core and marked by heightened expression of genes such as *EGFR* and *SOX9*. Further validation against independent datasets from healthy brain tissue and GBM bulk RNA-Seq reinforced the study’s findings. An intriguing observation was the conspicuous absence of astrocytes within the tumor core. Furthermore, a consistent expression profile for tumor cells in the peripheral zones was documented across all patient samples [60].

### 4.3. GRN Construction and Implementation

Our research transitions into its computational modeling phase. The central goal was to create a representative model of the GRNs to understand the dynamics of GBM’s subtypes and their inherent heterogeneity. In the subsequent section, we detail the methodology that forms the foundation of this computational framework.

#### Biological Criteria and Methodological Approach

The challenge we embraced was formulating a GRN that captures the essential dynamics behind GBM subtypes and heterogeneity. The GRN needs to be composed of regulatory interactions (edges) between the genes (vertices) of the system, which are supposed to be representative of the case under analysis (GBM). To begin this endeavor, the first step involved setting up clear biological criteria to guide the curation of these interactions. Our primary focus revolved around molecular mechanisms that directly or indirectly affect the number of mRNA molecules. Our approach embarked on an initial survey of direct and indirect interactions. *Direct interactions,* highlighted at the transcriptional level, are characterized by the binding of transcription factors (TF) to their target gene promoters. These interactions directly affect the amount of mRNA and are represented by a direct connection between the transcription factor vertex and the vertex representing the targeted gene. Conversely, *indirect interactions* encompass mechanisms that modulate the TF’s ability to bind to a gene promoter, such as the ubiquitination of a TF culminating in its degradation. Unlike direct interactions, the biological effects of indirect interactions are not immediate, and their consequences on the number of mRNAs need to be evaluated before they can be adequately represented within the model. As we delve into the complexities of these interactions, it becomes evident that a structured approach is needed to assemble the GRN.

To construct this survey of interactions, we began by pinpointing genes and markers pivotal for GBM subtypes [2]. The subtype classification was anchored in the schema presented by Verhaak et al. [21]. We then employed the MetaCore [61] (accessed on 16 April 2022) platform to search for interactions among these genes and markers. Specifically, we used the MetaCore transcription regulation network construction algorithm to build the network, identifying new vertices that bridge the initial gene list (provided in Appendix A). However, the output generated consisted of several disjoint subnetworks, with the initial genes scattered among them. We addressed this issue using the initial and added genes as a new input list. We repeated this until we obtained a connected network, ensuring that all the genes of interest were included in a single structure (Appendix A). With the GRN in place, the next step was to cross-reference it with the scRNA-seq expression data.

To achieve this alignment, an algorithm [62] in R [63] (version 4.1.2) was developed to match the scRNA-seq expression data with the network generated by MetaCore. The data preprocessing was performed using the Seurat package (version 4.1.1) [64], from which a *sctransform* normalization was applied to reduce technical bias and recover biologically significant distributions [50,65]. Cell cycle effects were not removed, as such information may lose its accuracy in tumor cells [42]. The algorithm then performed the following operations: (i) selected interactions classified as *Transcription Regulation*, (ii) intersected the genes of the network with those present in the data, and (iii) removed the genes that were associated with null values, even if they were present in the network. As a result, the network may change with variations in data, whether due to the selected cell types or patient IDs. With this adaptive network established, the subsequent step involved refining our working model to more accurately represent the cell types and their spatial localization within the GBM landscape.

Building upon our initial network, we adopted the cell type classifications proposed in Darmanis et al. [60]. These authors identified various cell types present in GBM samples, such as astrocytes, oligodendrocytes, and neurons, among others, through clustering and other methods. Centering our study on neoplastic cells nested within the tumor core, we delved into the dispersion of expression value for every patient consolidated in our dataset (Appendix A). To increase the probability of sampling over the entire phase space of GBM, we decided to obtain the landscape for all patients simultaneously instead of analyzing individual patients separately. This approach allows for better characterization of GBM by observing attractors related to the four subtypes of GBM while still reflecting the specific characteristics of each individual. By filtering for neoplastic cells located in the tumor core, we avoided incorporating the different features observed for neoplastic cells present in the periphery of the tumor. The filtration process culminated in an aggregate of 1027 cells, offering a robust foundation for downstream analyses.

To enable the automatic integration of the network with mathematical and computational models, we developed an algorithm available in the provided code repository [66], simplifying the process. The algorithm takes the network as input in a tabular format, representing the connectivity list between each vertex. It then processes the input and converts the tabular data structure into two directional graphs (digraphs), one for activation interactions and another for inhibition interactions, each represented by its adjacency matrix. With these transformed data structures, Equation (Equation 1) can be written as:(4)$$\begin{array}{cc} F=-kX & +\mathrm{rowsum}(M^{a}⊙V^{a})\\ \; & +\mathrm{rowsum}(M^{b}⊙V^{b})\;, \end{array}$$
where $k=\mathrm{diag}(k_{1},\ldots ,k_{N})$ is a diagonal matrix, $M^{a}$ is the activation matrix with entries ${\left(M^{a}\right)}_{ij}$ = $a_{ij}$, $M^{b}$ is the inhibition matrix with entries ${\left(M^{b}\right)}_{ij}=b_{ij}$, $V^{a}$ is the activation Hill functions matrix with entries:(5)$${\left(V^{a}\right)}_{ij}=\frac{X_{j}^{n_{ij}}}{S_{ij}^{n_{ij}}+X_{j}^{n_{ij}}},$$
and $V^{b}$ the inhibition Hill functions matrix with entries:(6)$${\left(V^{b}\right)}_{ij}=\frac{S_{ij}^{n_{ij}}}{S_{ij}^{n_{ij}}+X_{j}^{n_{ij}}}.$$

The ⊙ denotes the Hadamard product (element-wise matrix product) and rowsum(·) returns the vector with the row-wise sums of the matrix.

To observe the effects of perturbations, we represented the original adjacency matrices positions corresponding to activation (*a*), self-activation (sa), inhibition (*b*), and self-inhibition (sb) parameters as arbitrary values using symbolic computation. This method allowed further replacement of these symbols with numerical values. For example, a=sa=b=sb=1 during the parameter’s estimation, and a broader parameter space exploration for studying the basins’ stability. We could achieve the same by using multiplicative factors for each matrix element when i=j or i≠j.

### 4.4. Data Analysis: Dynamics Underlying Heterogeneity

Upon constructing our GRN, we have delineated the requisite genes and established the targeted scenario suitable for an in-depth investigation into GBM dynamics. The diverse clusters within our dataset underscore the heterogeneity of tumor evolution. This heterogeneity, especially when viewed through single-cell data, hints at the complex evolutionary trajectories of gene expressions.

At the heart of this exploration is the concept of the cancer attractor [18]. This paradigm posits that, despite genetic differences and the evident heterogeneity, cancer cells often gravitate towards a common state. It is this theoretical backbone that justifies our application of dynamic systems theory. Cellular states, acting as nonlinear evolving trajectories in a multidimensional space [13], are not merely random paths but rather can be viewed as potential courses directed towards these cancer attractors, converging at *basins of attraction.*

Given the challenge of real-time tracking, our focus shifts to snapshot-like data, which reflects cellular states at distinct temporal intervals. Such snapshots can be construed as potential distributions surrounding these cancer attractors for each GBM subtype. This interpretation allows studying system dynamics indirectly by observing the data variability [55]. Representations akin to Figure 1 enhance our understanding of the underlying dynamics and equip us with refined methodologies for data interpretation and parameter estimation. This integrated approach allows for a more nuanced perspective on the observed heterogeneity, paving the way for an optimized analytical framework.

In light of the cancer attractor concept, which underpins the dynamics evident in our Figure 1 hypothesis, the premise that ensemble and time averages converge becomes central. For example, in the case of aggressive malignancies, such as GBM, mutation accumulation and disease progression might lead to a swifter exploration of phase space. It is imperative to note that not all cells are destined to traverse every available state over extended time intervals. Our hypothesis applies to the system’s intrinsic components (‘basins of attraction’), as delineated in Palmer [44] and expanded upon in later studies, such as [45,67]. Given GBM’s inherent aggressiveness, it is posited that the heterogeneity discerned from our spatial samples provides a panoramic view of this attractor’s topography. The cellular subsets within the spatial samples are believed to span a substantial segment of the attractor.

To initiate, we discern clusters within the data (refer to Section 4.2), considering the centroids of these clusters as approximations of equilibrium states. With this presumption, the driving forces (i.e., expression level change rates) near cluster centroids are assumed to be negligible. Consequently, the system can be characterized by:(7)$$F=\frac{dX}{dt}\approx 0\;,$$
which enables the system’s stability characterization through parameter estimation (see Section 4.6.2).

Our characterization of clusters relies on dimensions associated with four GBM subtype markers: *EGFR* (classic), *IDH1* (proneural), *NEFL* (neural), and *CD44* (mesenchymal) [2]. Unlike genes linked with the cell cycle or prone to high variability, markers must exhibit constrained expression ranges, ensuring clustering and landscape visualization reliability. While many clusters likely represent distinct cell types and subtypes, others may be spurious due to artifacts, noise, or external factors. Some clusters could also reflect metastable states (short-lived configurations within the phase space). This necessitates careful analysis and validation when interpreting clustering outcomes within the GRN framework.

To operationalize the aforementioned considerations, we employed Mathematica (Version 13.1) [68] to analyze the gene expression data from Darmanis et al. [60]. We applied a dimensionality reduction (t-SNE) to identify the clusters, followed by two clustering methods, k-means and Neighborhood Contraction (NbC). K-means is a popular and computationally efficient algorithm partitioning data into spherical groups [51]. NbC is a density-based method that identifies clusters of varying shapes and densities without a prior cluster number definition [52].

### 4.5. Mapping the Landscape: Subtypes Distributions

Given GBM’s complex landscape, it is crucial to define the framework for our study. This landscape can be visualized as a hierarchical structure, where each layer or ‘envelope’ captures different cellular dynamics. At a high level, the landscape encapsulates cell-type attractors. Delving deeper, it reveals the intricacies of dynamics within cell types, highlighting subtypes. An even more granular approach would reflect processes within these subtypes, such as metabolism.

Our analysis targets the intermediary level—that of dynamics among subtypes, situated within the broader GBM ‘basin of attraction’. This choice enables us to provide detailed insights while maintaining a manageable computational scope. Through our modeling, distinct basins of attraction have been identified, emphasizing GBM’s inherent heterogeneity. These basins reflect genomic instability, epigenetic regulations, and selective pressures driving the observed diversity.

We applied the central limit theorem and the law of large numbers, considering the centroids of the clusters as means of Gaussian distributions and representing gene expression levels associated with potential GBM subtypes. Additionally, we computed and stored each cluster’s standard deviation based on the respective averages, alongside the proportions of cells they contain.

Therefore, we modeled gene expression as a Gaussian distribution given by:(8)$$P\left(X\right)=\frac{1}{\sqrt{2\pi \sigma^{2}}}\exp\left[-\frac{{\left(X-\bar{X}\right)}^{2}}{2\sigma^{2}}\right]\;,$$
where $\bar{X}$ represents the sample mean and σ represents the (unbiased) sample standard deviation.

Gene interactions exhibit intricate dynamics, with alterations in one gene influencing others. Nevertheless, in steady-state systems, these interactions are reflected in the statistical distribution, enabling independent consideration of each gene (mean-field approximation). By focusing on the moments that depict the resultant distributions from the system’s collective interactions, we can manage a more tractable analysis, capturing the essential behavior of the stochastic dynamics.

The mean-field approximation helps to compute the Waddington (epi)genetic landscape, enabling the study of state transitions and the probability of observing specific gene expression distributions. This approximation allows each attractor α probability to be described by [37]:(9)$$P_{\alpha }\left(X_{1},\ldots ,X_{N}\right)=\prod_{i=1}^{N}P_{\alpha ,i}\left(X_{i}\right)\;,$$
where *i* is the index of genes, *N* is the number of genes, and $P_{\alpha ,i}$ is defined as (Equation 8), each one with sample mean ${\bar{X}}_{i}$ and sample standard deviation $\sigma_{i}$.

The probability of a cellular state will be given by the steady-state probability $P_{ss}$ [37]:(10)$$P_{ss}=\sum_{\alpha =1}^{N_{clusters}}w_{\alpha }P_{\alpha }\;,$$
where $w_{\alpha }$ is the percentage of cells in each attractor and $N_{clusters}$ is the maximum number of attractors (clusters) found by the method.

The total steady-state probability (Equation (Equation 10)) can be estimated for experimental by clustering the scRNA-seq data and for the simulated data by allowing sufficient time for the in silico system to evolve from sampled initial conditions towards steady states. Once we obtain the steady-states, the process is the same as for scRNA-seq data.

Computing steady-state probabilities enables the system’s global behavior to be studied using (epi)genetic landscapes. This theory is based on the flow theory of nonequilibrium dynamical systems, as discussed in Wang [69]. Obtaining steady-state probability $P_{ss}$ through sampling rather than an analytical solution of the Fokker–Planck equations leads to the *populational landscape.* This landscape *U* is defined as derived from the negative logarithm of the total probability of the system [37]:(11)$$U=-\ln P_{ss}\;,$$
where *U* is a dimensionless potential that quantifies the probabilities of states and their transitions. Higher probability states correspond to lower potentials (greater depths), while the barriers between the basins of attraction are related to the time spent in each state.

In cases where the system is in equilibrium, with detailed balance preserved, the Boltzmann relation holds. For these cases, the landscape corresponds to the equilibrium probability. However, in cases out of equilibrium, as considered here, rates of transitions between states do not balance each other. This causes trajectories not to follow the gradient but to be characterized by a probability flux, as discussed in Wang et al. [34].

Despite these complexities, the (epi)genetic landscape can still provide valuable insights into the behavior of systems out of equilibrium. By investigating the nonequilibrium (epi)genetic landscape, we expect to gain insights into the impact of stochastic fluctuations and transient states. This can help guide further investigations and develop more accurate models for the system’s behavior.

### 4.6. In Silico Dynamics and Simulated Landscape

Having delineated the intricate (epi)genetic landscape of GBM data, we transition to constructing an in silico dynamical model. This endeavor seeks to simulate a compatible landscape and evaluate the emergent properties of the in silico system compared to our initial observations. Central to this modeling process is the accurate representation of gene interactions within the GRN. To achieve this, we employ regulation functions, with a particular emphasis on Hill functions.

#### 4.6.1. Specifying Dynamics: Regulation Functions

Hill functions are widely used for modeling GRN interactions, accounting for the influence of activators and inhibitors on regulation strength. Their steepness reflects the gene expression’s sensitivity to variations in regulatory molecule concentrations. They are typically characterized by constant coefficients for each interaction, resulting in a uniform regulation strength when applied to all target genes of the same TF. As monotonic functions, they exhibit increasing regulation strength in response to increasing TF concentrations. These features can lead to obstacles in modeling cancer heterogeneity.

Cancer heterogeneity implies cells might present different regulations. In other words, the same TF concentration might lead to different regulation strengths for different cells. For example, one gene may exhibit high expression for a specific number of TF molecules while another presents low expression. Typical features of Hill functions cannot model this dependence on the target gene.

Another important aspect is that they might face challenges modeling complex regulatory mechanisms when the network is incomplete. For example, gene expression levels are typically biologically constrained. This could be due to internal regulatory interaction or even microenvironment responses. In any case, both constraints are typically intricate to implement [70]. One reason is the need to know a priori all interactions that could constrain gene expression levels. Another is that even knowing these constraints, it would still need to implement a very complex model with complicated network dynamics.

To overcome these limitations, we introduced modified Hill functions, which provided a flexible framework for modeling the relationships between genes and regulators, allowing experimental data to constrain the dynamics. We proposed the modified Hill functions given by:(12)$${\left(V^{a}\right)}_{ij}=\left(\frac{X_{j}^{n}}{{\left(h\left(X_{i}\right)+S_{j}\right)}^{n}+X_{j}^{n}}\right)\;,$$
for activation, in place of (Equation 5), and by:(13)$${\left(V^{b}\right)}_{ij}=\left(\frac{S_{j}^{n}}{{\left(h\left(X_{i}\right)+S_{j}\right)}^{n}+X_{j}^{n}}\right)\;,$$
for inhibition, in place of (Equation 6). The index *i* is for the vertex related to the target, *j* for the vertex affecting the target, and *h* is a modifier function used to obtain shapes with desired properties.

We tested several options for *h*, including the original model with h(x)=0 to determine the best model. The proposed possibilities were:(14)$$\begin{array}{cc} h_{1}\left(x\right) & =x\sqrt{\prod_{\alpha }\left|x-{\bar{x}}_{\alpha }\right|}\;, \end{array}$$
(15)$$\begin{array}{cc} h_{2}\left(x\right) & =\left(\frac{1}{0.1x+0.001}\right)+\sqrt{\prod_{\alpha }\left|x-{\bar{x}}_{\alpha }\right|}\;, \end{array}$$
(16)$$\begin{array}{cc} h_{3}\left(x\right) & =\left(\frac{1}{0.1x+0.001}\right).\sqrt{\prod_{\alpha }\left|x-{\bar{x}}_{\alpha }\right|}\;, \end{array}$$
where ${\bar{x}}_{\alpha }$ is the gene average considering each attractor α. The first term of Equations (Equation 15) and (Equation 16) was chosen to ensure appropriate behavior for values close to zero with the constants in the denominator empirically verified to optimize the results.

In addition, we considered:(17)$$S_{j}=f\cdot \left(\max_{\alpha }\;\left[X_{j}^{\alpha }\right]+1\right)\;,$$
where *f* is a proportionality constant and $\max_{\alpha }\;\left[X_{j}^{\alpha }\right]$ represents the transcription factor with the highest expression value among all attractors. Adding a unit to the base value of *S* ensures that the proportionality constant can adjust cases with only zeroes and avoids null denominators. This modification sets the value as a fraction of the maximum gene expression for all attractors found. This assumption is a biological simplification, proposing that regulation intensity is proportional to the highest equilibrium value of the transcription factor. This approach helps capture the effects of weaker transcription factor levels when using a single sigmoid function to represent regulation intensity. Lower levels could rapidly increase the sigmoid function from zero, impeding accurate modeling of regulation intensity across various transcription factor concentrations.

These modifications were proposed to impose constraints that are difficult to add to the system using a graph structure alone. By introducing these changes, we aim to make the model less sensitive to the incompleteness of the gene regulatory network and better integrate hidden information within the data to improve the biological description of the system dynamics. Additionally, these modifications allow us to consider different regulation intensities for each network interaction, which is a more realistic representation than assuming the same value for all interactions.

#### 4.6.2. GRN Edges and Parameter Estimation

Within the context of GRN and cancer, it is observed that parameter values do more than modulate gene interactions. When certain values approach near-zero levels, they can alter the GRN topology, removing specific network connections altogether. This reflects the potential for cancer to reconfigure cellular regulatory landscapes and allows the model to indirectly capture GRN topology from scRNA-seq data in the parameter estimation process.

Dynamic model parameters can be categorized into deterministic and noise-related parameters. These parameters must be able to retrieve expression values compatible with the data and known biological behaviors. Since the expression values obtained from scRNA-seq data result from genetic and epigenetic regulation, using these data for parameter estimation requires implicitly considering all contributions, including those from the microenvironment.

It is important to note that the parameters are estimated for the system at values close to equilibrium. There is no guarantee that they will remain consistent in regions far from equilibrium. States far from equilibrium might exhibit genetic and epigenetic differences, deviating from the behavior predicted by the parameter tuning. Nevertheless, the dispersion of experimental data described by the model may sufficiently reflect the statistical dynamics of GBM for characterizing its subtypes.

We proposed different ways of addressing the challenges associated with estimating the deterministic parameters in our model. These challenges arise from the number of parameters, the selection of an appropriate estimation method, and the need to ensure biological interpretability. One concern when dealing with a large number of parameters is overfitting; thus, it is important to be aware of this issue and to approach it cautiously.

Additionally, identifiability is crucial, as selecting unique parameters estimated from the available data can be difficult. This challenge may lead to multiple sets of parameter values that produce similar model outputs, making it difficult to draw meaningful conclusions. Furthermore, estimating many parameters often requires significant computational resources and time due to the increasing search space for the parameter values. The complexity of models with many parameters can make them harder to interpret and understand, as the relationships between variables may be obscured.

Considering these factors, we explored various parameter estimation strategies that balance model complexity, computational efficiency, and biological interpretability. We considered the following three scenarios: (i) with two parameters per Equation (one for activation and one for inhibition); (ii) with one parameter related to each input vertex (divided between activation and inhibition), that is, 2n parameters per Equation (including null values); and (iii) with a combination of (i) and (ii), i.e., n×n parameters (including null values).

Departing from Equation (Equation 7), Equation (Equation 18) illustrates the first case:(18)$$k_{i}X_{i}=a_{i}\left(\sum_{j}V_{ij}^{a}\right)+b_{i}\left(\sum_{j}V_{ij}^{b}\right)\;,$$
where a possible biological interpretation is an activation and inhibition intensity proportional to the target gene, for example, due to epigenetic regulations. Equation (Equation 19) brings the second case:(19)$$k_{i}X_{i}=\left(\sum_{j}a_{j}V_{ij}^{a}\right)+\left(\sum_{j}b_{j}V_{ij}^{b}\right)\;,$$
where the consideration of 2n parameters would be due to intermediate factors affecting the interactions preceding the binding to the promoter region and the resulting gene transcription. Equation (Equation 20) brings the last case:(20)$$k_{i}X_{i}=a_{i}\left(\sum_{j}a_{j}V_{ij}^{a}\right)+b_{i}\left(\sum_{j}b_{j}V_{ij}^{b}\right)\;,$$
where the idea was (i) to obtain a different parameter for each edge and (ii) to capture as much information as possible. Each activation coefficient would be $a_{ij}=a_{i}a_{j}$, with an equivalent procedure for the inhibition coefficients $b_{ij}$.

To estimate the parameters of cases (i) and (ii), we used the $L_{1}$-norm robust regression that can be solved as a linear programming problem [71]. We used the Simplex algorithm in the Mathematica environment [68]. Assuming uniform and constant degradation coefficients for all mRNA molecules, we have for all gene *i*, $k_{i}=k$, and Equations (Equation 18) and (Equation 19) can be rewritten in the form of the following equation:(21)kX=Vc,
where $V=(V^{a}\;|\;V^{b})$, $c=(c^{a}\;|\;c^{b})$, for ${\left(c^{a}\right)}_{i}=a_{i}$ and ${\left(c^{b}\right)}_{i}=b_{i}$.

The parameter estimation was conducted by simultaneously incorporating all centroids of all basins of attraction, meaning that the parameters were chosen to capture the contributions of all possible equilibrium states of the system. This approach avoided overfitting individual clusters, potentially hindering other attractors’ representation. Mathematically, for each centroid vector $X_{\alpha }$, we build the matrices $V_{\alpha }$ and the vectors $\beta_{\alpha }=kX_{\alpha }$, and stack them as:(22)$$\begin{array}{cc} M & =[V_{1}\;|\;\cdots \;|\;V_{N_{clusters}}{]}^{T}, \end{array}$$(23)$$\begin{array}{cc} \beta & =[\beta_{1}\;|\;\cdots \;|\;\beta_{N_{clusters}}{]}^{T}. \end{array}$$

By doing so, we solve the $L_{1}$-norm minimization problem:(24)$$\begin{array}{cc} \min\; & ||Mc-{\beta ||}_{1}\;, \end{array}$$(25)$$\begin{array}{cc} \; & c\geq 0\;, \end{array}$$
then we compute and store the maximum residual $R_{\infty }=||Mc-{\beta ||}_{\infty }$ and the total $R_{1}=||Mc-{\beta ||}_{1}$ for each fit as a regression quality measurement.

We obtain the solutions for cases (i) and (ii) for combinations of *n* and *S*, the regulation functions coefficients. The values of *n* ranged from 1 to 4 in increments of 1, while the proportionality constants $f_{a}$ and $f_{b}$ ranged from 0.1 to 1.3 in increments of 0.2. These constants were applied separately for activation and inhibition regulation functions. We used each of the proposed regulation functions for each parameter combination. After estimating the parameters, the absolute residuals list, $R_{\infty }$, and $R_{1}$ were saved. We performed another estimation for each set of parameters obtained using Equation (Equation 19). This additional estimation allowed us to combine the newly estimated parameters with the initial set, as in Equation (Equation 20).

#### 4.6.3. Noise Characterization and Stochastic Simulation

We subsequently solved Equation (Equation 3) numerically using the Euler–Maruyama method with an Itô interpretation, where noise is added before increasing expression levels. A Stratonovich interpretation is also possible, as discussed in [56]. We employed a multiplicative noise function g(X,t), which accounts for possible mean and standard deviation dependence after a logarithm transformation [49,50], as shown in Equation (Equation 26):(26)g(x;c0,c1,p)=xc0x+c1−1x−1px(1−p)c1.
where $c_{0}$ is the noise amplitude that scales a negative binomial distribution, with $c_{1}$ and *p* empirically determined to fit the data best, *p* is any positive real number less than or equal to 1, and *x* represents the expression level as any positive real number.

For initial conditions such as experimental data points or attractor coordinates, the system is expected to evolve towards the centroid of the cluster it belongs to or maintain trajectories around its average value. We performed stochastic simulations to investigate the system dynamics with a time interval of 50 a.u. (time steps of Δt=0.1) to ensure the system reaches equilibrium states. We varied the noise amplitude $c_{0}$ (3.5 and 7.0) and explored the impact of different activation and inhibition levels (*a*, sa, *b*, and sb) from 0.6 to 1.4 (0.1 by 0.1). These settings were determined through extensive preliminary simulations to understand the system dynamics better, reproduce the observed variability in the experimental data, and maintain computational feasibility.

We then stored the parameter configurations that exhibited an average of 15 or more genes remaining within two standard deviations from the respective attractors’ values (as identified by clustering and fitted with the parameters to be stable points) as described in Equation (Equation 27). In essence, given:(27)$$z=\frac{|\bar{x}-\mu |}{\sigma }\;,$$
where μ represents the cluster centroid of each gene, $\bar{x}$ denotes the time average within an interval from time 30 to 50 (allowing the system to fluctuate around the centroid), and σ is the standard deviation of the simulated attractor. If z≤2 for at least 15 genes out of the 40-gene vector, we stored the parameter set. This approach allowed us to test the hypothesis that the clusters found in the data are statistically significant compared to those obtained by the model with the determined set of parameters. The initial condition was sampled from a uniform distribution for computing the landscapes. This sampling method was employed on the notion that the trajectories of the system, given sufficient time, would populate regions of the phase space in a manner consistent with their statistical significance.

#### 4.6.4. Clusters Comparison and Simulated Landscape

At the culmination of our previous section, we underscored our strategy of analyzing the in silico data akin to the experimental scRNA-seq data. This involved utilizing cluster centroids as a close approximation of fixed points to delineate the attractors and their respective basins. Such an approach serves as an analytical framework and paves the way for the critical validation step: a juxtaposition of in silico and experimental data distributions. To what extent could we reproduce data centroids and heterogeneity considering stochastic fixed point dynamics? Given the necessity of this validation, an essential preliminary step was undertaken—endeavoring to align simulated clusters with experimental ones.

To facilitate the clustering comparison, we rearranged each simulated data cluster to match its closest experimental data cluster. The rearranging starts with (i) computing the centroids of experimental and simulated clustered data. Both cases contain grouped cells linked to each cluster stored in a list of lists manner, with each sublist representing the cells of an α cluster. This way, the centroids for each case are stored in an α-ordered list. With two lists of centroids, we then (ii) calculated the Euclidean distances between each element using the experimental centroids as a reference. This step ends up creating a distance matrix. (iii) The reordering starts with an algorithm that identifies the simulated centroid corresponding to the smallest distance to one of the experimental ones. (iv) These indices are mapped together and removed from the distance matrix. This step ensures that each element in the simulated set is associated with a unique experimental cluster. (v) This process is repeated until all experimental centroids are mapped to one of the simulated data. (vi) The code returns the reordered simulated clusters in a list of lists format. (vii) For differing numbers of clusters, the ones not matched are appended at the end. This code pseudocode is presented in Algorithm 1. This process aids in identifying similarities and differences between the two sets and helps to understand the underlying structure of the data.
**Algorithm 1** Reorder Clusters**Require:** *clusters1*, *clusters2***Ensure:** Combination of sorted and remaining clusters1:*centroids1* ← computeCentroids(*clusters1*)2:*centroids2* ← computeCentroids(*clusters2*)3:**for all** *centroid1* in *centroids1*
**do**4:    Calculate the distance to each *centroid2* in *centroids2*5:**end for**6:Find the smallest distance and corresponding *clusters* indices7:Create new ordering of *clusters2* based on these indices mapping8:*sortedClusters* ← Newly ordered *clusters2*9:*remainingClusters* ← *clusters* in *clusters2* not in *sortedClusters*10:*combinedClusters* ← Combine *sortedClusters* with *remainingClusters* appended at the end11:**return** *combinedClusters*

Finally, we use different ways to assess the compatibility between experimental and simulated distributions, including their respective landscapes, computed using Equations (Equation 9) to (Equation 11).

### 4.7. Dynamics Inside Basins of Attraction

Within the framework of our model, we sought to understand the depth of GBM’s heterogeneity by examining the basins of attraction. Our primary focus was to assess the model’s capability to capture data heterogeneity evident in GBM. We aimed to ascertain the stability and scope of the cancer subtype attractors.

Additionally, we ventured into the statistical dynamics inherent within each attractor. This involved evaluating the consistency between sample and time averages, a vital aspect of the used framework. We employed autocorrelation functions, which facilitated the determination of the system’s inherent timescales and the propensity for transitions between attractors.

#### 4.7.1. Gene Expression Dispersion in Cancer

The dispersion observed in cancer data prompts inquiries about the underlying dynamics causing such patterns [72]. In this study, we postulate that stochasticity, chaotic dynamics, or a combination of both might influence the observed dispersion. Regardless of the underlying reason, we assume these dynamics unfold within specific basins of attraction (see Appendix B).

We view gene expression dispersion as a consequence of cancer progression. Genetic mutations, epigenetic shifts, and changing cellular microenvironments amplify the system’s diversity, resulting in more intricate and varied expression distributions. Such diversity mirrors the myriad ways cancerous cells can organize and interact, mirroring the increasing complexity and heterogeneity of the tumor environment [73].

Past research has delved into the rising entropy associated with these phenomena [74,75]. This increase in entropy points to varied cellular states and possibly fluctuating energy levels. Given that differences in energy levels between cell types closely intertwine with genetics and epigenetics, our aim is to examine the stability of cancer subtypes through the lens of these varied states’ attraction basins. We utilize the Waddington landscape concept to study the system’s behavior upon settling into these configurations instead of focusing on the journey leading to them.

Our approach centers on observing the system’s properties over time to comprehensively capture its dynamics and fluctuations. Moreover, sampling must span a broad spectrum of potential states. We conceive each basin of attraction as a distinct subsystem, wherein the movement between them is minimal yet possible. Such a perspective aligns with the notion of attractors and the definition of subtypes; frequent transitions between subtypes would undermine their distinct categorization.

Furthermore, rapidly evolving malignancies, such as GBM, are expected to explore the phase space more extensively than healthy cells’ constrained dynamics or less aggressive tumors. Our analysis, thus, focuses primarily on these basins of attraction, adding rigidity to our suppositions.

#### 4.7.2. Fixed Points: Sample vs. Time Averages

In dynamical systems modeling GBM, attractors represent stable cellular states the system naturally gravitates towards. These cancer attractors, identified through cluster centroids, elucidate the system’s tendencies. When analyzing variations around these attractors, the time average and sample average become crucial. Ideally, the time average should align with the sample average from various trajectories for a system closely orbiting an attractor. This alignment sheds light on GBM’s inherent dynamics and the stability of its states.

To elucidate this phenomenon mathematically, let us contextualize our variables. The expression level of gene *i* at a particular time point within the interval *T* is denoted by $X_{i}\left(t\right)$. The spatial domain or basin, in which these expression levels predominantly lie and which is closely associated with the specific cancer attractor A, is termed B(A) (refer to Appendix B for a comprehensive definition). Within this schema, we propose a model suggesting that the alignment between the time and sample averages becomes increasingly robust as the temporal subinterval $T_{12}=t_{2}-t_{1}$ enlarges and the sample size, *N*, increases. This alignment resonates with our understanding of GBM’s aggressive progression and manifestation over different timescales. The underlying principle is straightforward: our mathematical approximation is closer to empirical reality as we gather more data samples and witness increased cellular aggression (indicating shorter times to span the associated basins). This time average expression for gene *i* can then be expressed as ${\langle X_{i}\rangle }_{T_{12}}$, given by:(28)$${\langle X_{i}\rangle }_{T_{12}}=\frac{1}{T_{12}}\int_{t_{1}}^{t_{2}}X_{i}\left(t\right)dt$$
with the time average of $x_{i}$ in the subinterval $\left[t_{1},t_{2}\right]$, and the sample average:(29)$${\langle X_{i}\rangle }_{A}=\frac{1}{N_{samples}}\sum_{j=1}^{N_{samples}}{\left(X_{i}\right)}_{j},$$
where $N_{samples}$ is the sample size in the basin of attraction B(A) and ${\left(X_{i}\right)}_{j}$ represents the *j*-th sample of the gene expression level $X_{i}$ within the basin of attraction B(A).

Our foundational assumption rests on the idea that:(30)$$\lim_{T_{12}\to \infty ,N\to \infty }\left|{\langle X_{i}\rangle }_{T_{12}}-{\langle X_{i}\rangle }_{A}\right|=0,$$
where $T_{12}$ is the length of the subinterval and the sample size *N* approaches infinity; the difference between the time and space average approaches zero.

#### 4.7.3. Timescales and the Propensity for Transitions

Finally, timescale separation helps assess the relaxation time of $X_{i}$ within B(A) and outside of it. We propose this by examining the decay rates of the autocorrelation function for $X_{i}\left(t\right)$. The autocorrelation function measures the similarity between a variable and its lagged version, with a faster decay implying a shorter relaxation time. This behavior would suggest a propensity for reaching equilibrium. The autocorrelation function of $X_{i}$ is given by [76]:(31)$$r_{X_{i}X_{i}}\left(\delta \right)=\frac{\sum_{n}^{T_{max}-\delta }\left(X_{i,n}-\bar{X}\right)\left(X_{i,n+\delta }-\bar{X}\right)}{\sum_{n}^{T_{max}}{\left(X_{i,n}-\bar{X}\right)}^{2}}$$
where $r_{X_{i}X_{i}}\left(\delta \right)$ is the autocorrelation at lag δ, $X_{i,n}\approx \;X_{i}\left(t_{n}\right)$ is the value of the time series at time step *n* ($t_{n}=n\Delta t$, where Δt is the time step interval), $X_{i,n+\delta }\approx \;X_{i}\left(t_{n}+\delta \right)$ is the value of the time series at time step n+δ, $\bar{X}$ is the mean of the time series, and $T_{max}$ is the total number time series points. The autocorrelation ratio normalizes the measure to stay between −1 and 1.

The autocorrelation function provides valuable insights into the system dynamic guided by a driving force. Derivatives greater than zero imply a system driven by network interactions. In this case, the autocorrelation values at different lags capture a pattern and may not oscillate around zero. As the lag increases, the autocorrelation values might decrease, but still with the influence of the driving force.

On the other hand, the driving force’s influence becomes constant or negligible for a system in a steady state. Being mainly affected by stochastic fluctuations, the autocorrelation at different lags likely oscillates around zero. It indicates little or no correlation between points and more unpredictable behavior, less dependent on past values. We expect to achieve this behavior fast once centroids (i.e., attractors) are used as starting points. In this case, discrepancies would possibly relate to nonstability or transitions.

To proceed with the autocorrelation analysis, we defined a vector K with each element representing the autocorrelation at a specific lag δ. Thus, K can be computed as follows:(32)$$K=\left[r_{X_{i}X_{i}}\left(1\right),r_{X_{i}X_{i}}\left(2\right),\ldots ,r_{X_{i}X_{i}}\left(T_{max}-1\right)\right]$$
with each value $r_{X_{i}X_{i}}\left(\delta \right)$ computed using Equation (Equation 31). By computing the autocorrelation values for increasing lags, we measure if time series values that are further apart are linearly related or correlated.

To quantify the relaxation time, we defined the timescale by:(33)$$\delta^{*}=\min\delta |K<e^{-1}$$
where $\delta^{*}$ is the lowest lag δ, related to a characteristic timescale, at which the autocorrelation value falls below the threshold $e^{-1}$.

In conclusion, these methods offer a strategy to validate the aforementioned criteria, delving into the consistent gene expression patterns within the simulated dataset. This examination underscores how such assumptions can significantly enhance our understanding of GBM GRN dynamics derived from scRNA-seq data.

## 5. Conclusions

Despite advances in the analysis and availability of scRNA-seq data, the dynamics of GRNs are still primarily investigated using arbitrary parameter values. Our work goes beyond descriptive and analytical methods, proposing a conceptual and methodological investigation using scRNA-seq to interpret the heterogeneity and quantify parameter values related to the dynamics in the context of GBM. By investigating statistical properties of GBM aggressiveness, e.g., cluster centroids as fixed points describing the cancer attractor, our method connected biological features (heterogeneity and aggressiveness) and statistical properties of the stochastic dynamics of a GRN model. We proposed that the aggressive nature of GBM influences the dynamics and determines how extensively the cancer attractor is sampled. In this context, the sample average (used to identify the cluster centroid) is equivalent to the time average of a representative sample traversing the cancer attractor for an extended duration.

Our findings demonstrated compatibility between experimental and simulated data centroids. We also investigated the stability and transitions between attractors. Despite the model simplifications, our results offered insights into the dynamics of GBM. The transitions between the basins revealed a possible interplay between subtypes, potentially uncovering factors that drive cancer recurrence and progression. Further exploring the dynamics within the (epi)genetic landscape of GBM subtypes can help understand the path leading to the differentiation of each subtype. This investigation might help uncover dynamical principles underlying cancer development and correlate with molecular mechanisms. We expect that determining unique molecular mechanisms related to the statistical properties of cancer heterogeneity might help develop potential diagnostic tools and personalized therapeutic interventions. Finally, connecting the ’geometry of heterogeneity’ with instability mechanisms could offer a different perspective on tumor biology. This approach could be used to assess and monitor the evolution of malignant states, serving as an instrument for diagnosis and treatment.

## Figures and Tables

**Figure 1 ijms-25-04894-f001:**
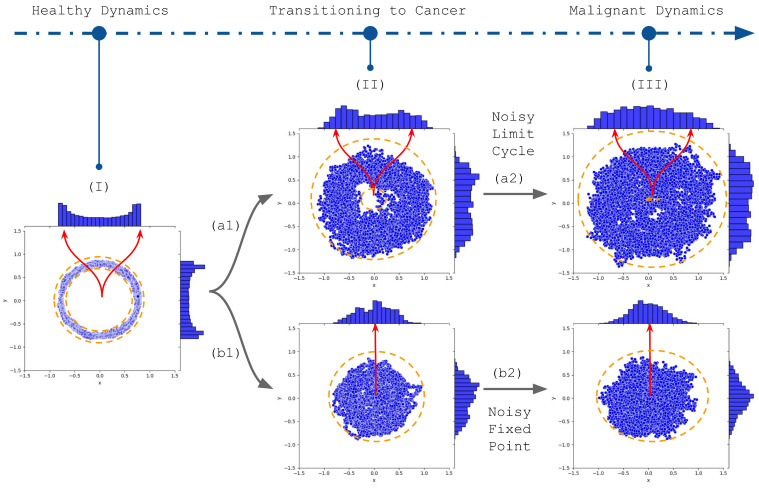
This image depicts our hypothesis about a cellular transition from a ’healthy’ limit cycle to a malignant state, a transformation that would enlighten the dynamics underlying the single-cell RNA sequencing data heterogeneity. The ‘initial state’ (**I**) represents a stable cellular trajectory. However, it is important to note certain dimensions—potentially related to marker genes— might already display a fixed-point dynamic. The outcomes of genetic, epigenetic, and microenvironmental alterations are depicted through directional arrows (a1 to a2, and b1 to b2). The upward trajectory (a1 to a2) underscores an expanded oscillatory boundary called a stochastic tube. Notably, the initial limit cycle distribution (**I**) is characterized by two noncentral peaks, which witness a reduced prominence due to augmented stochastic fluctuations (**II**), resulting in an irregular distribution resembling oscillations around a stable point (**III**). In contrast, the descending trajectory (b1 to b2) emphasizes an amplified fluctuation envelope coupled with parameter shifts associated with a Hopf bifurcation, consolidating the dynamics of a malignancy state around a fixed point. The fixed point histogram displays a peak aligned with the fixed points, highlighting the relevance of features such as clustering centroids in capturing the nuanced hallmarks of malignant transformation.

**Figure 2 ijms-25-04894-f002:**
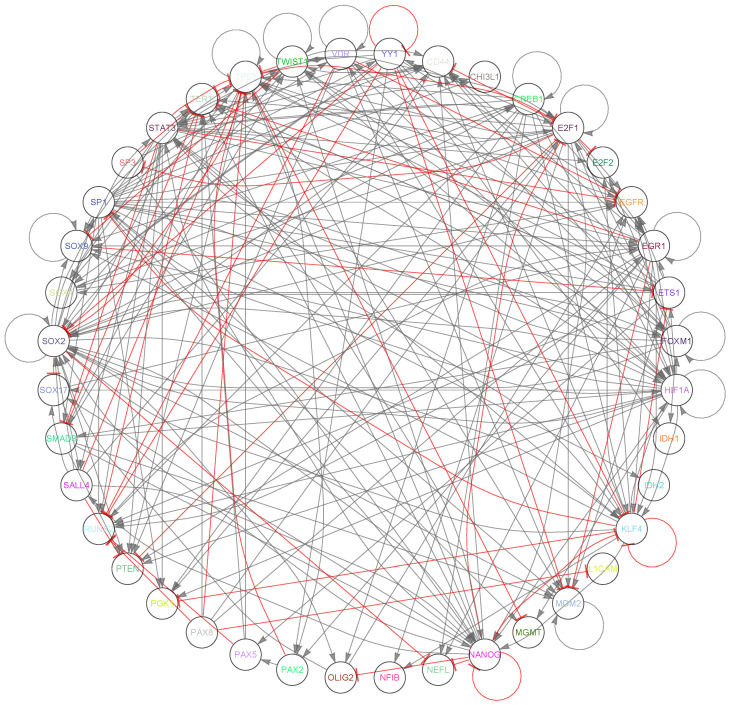
Gene regulatory network for single-cell RNA sequencing of the tumor core of four patients with GBM. Grey lines with flat arrows represent activations, and red lines with arrowheads represent inhibitions. It contains 40 vertices and 242 edges, with 187 activations, 11 self-activations, 41 inhibitions, and 3 self-inhibitions.

**Figure 3 ijms-25-04894-f003:**
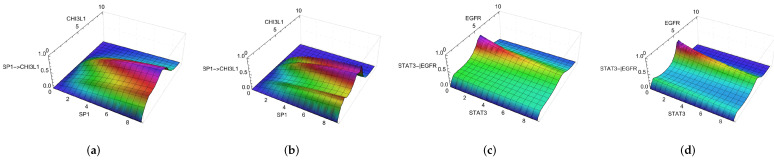
New regulation function *V* with n=1 and $h_{2}\left(x\right)$ for different combinations of genes and/or transcription factors. The horizontal axis represents the transcription factor and gene quantification using the normalized amount of single-cell RNA sequencing of experimental data. The vertical axis represents the quantification of the interaction regulations. The colors represent the vertical axis values, highlighting the representation of the 3D dimension. (**a**,**b**) Activation values using the k-mean and NbC clusters, respectively, ($f_{a}=0.1$); (**c**,**d**) Inhibitory interactions using the k-mean and NbC clusters, respectively, ($f_{b}=1.3$ and $f_{b}=1.1$).

**Figure 4 ijms-25-04894-f004:**
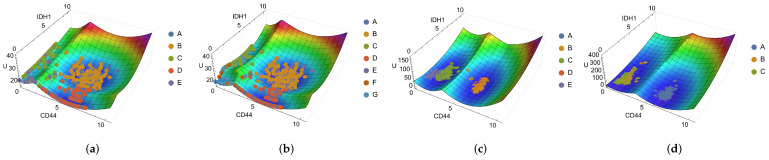
(Epi)genetic landscapes for experimental (**a**,**b**) and simulated (**c**,**d**) data, with experimental and simulated points overlaid for compatibility visualization. (**a**) Landscape for experimental data with k-means clusters; (**b**) landscape for experimental data with NbC clusters; (**c**) landscape for simulated data after parameter estimation with the k-means centroids and clustering with Gaussian Mixture; (**d**) landscape for simulated data after parameter estimation with the NbC centroids and clustering with Gaussian Mixture. The horizontal axes show the expression values of each marker gene, while the vertical axis represents the values of the landscape.

**Figure 5 ijms-25-04894-f005:**
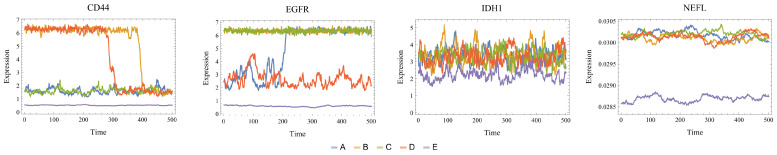
Trajectory plots for each basin (A, B, C, D, and E), showcasing the dynamics of the four marker genes (CD44, EGFR, IDH1, and NEFL). The trajectories illustrate the time evolution from initial conditions chosen as the centroid of the clusters found by the Gaussian mixture algorithm and using the parameters estimated from the k-means centroids. The horizontal axis represents time steps, and the vertical axis shows the expression level of each gene.

**Figure 6 ijms-25-04894-f006:**
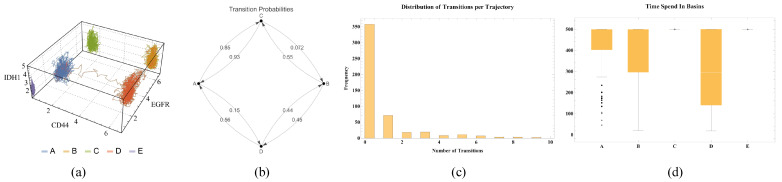
(**a**) Three-dimensional visualization of full trajectories in three of the markers space. All axes display the expression values of each marker gene. Each line represents an entire time of the three considered trajectories. Each color/letter indicates its respective basin. (**b**) Transition graph illustrating the connections between different basins. Vertices represent basins, and edge weights represent the probabilities of transitions between basins. (**c**) Frequency of transitions per trajectory. Histogram showing the frequency of transitions between basins in each trajectory, highlighting that most trajectories do not present any transition, and those that do tend to have a small number of transitions. The vertical axis shows the frequency of each number of transitions per trajectory, while the horizontal axis shows the number of transitions per trajectory. (**d**) Analysis of time spent in basins before a transition. Box plots reflect the distribution of time spent in each basin across all trajectories before they present a transition. The vertical axis represents the time spent in the basin, while the horizontal axis represents the correspondent basins.

**Figure 7 ijms-25-04894-f007:**
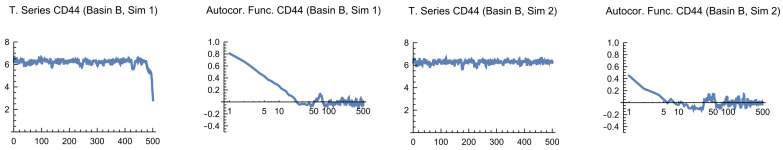
Autocorrelation analysis of time series data for CD44, basin B, and two simulations. Each pair of plots within includes a time series plot (**left**), with the horizontal axis representing time and the vertical axis representing expression values, and an autocorrelation plot (**right**) with the horizontal axis representing time lags and the vertical axis representing autocorrelation values.

**Figure 8 ijms-25-04894-f008:**
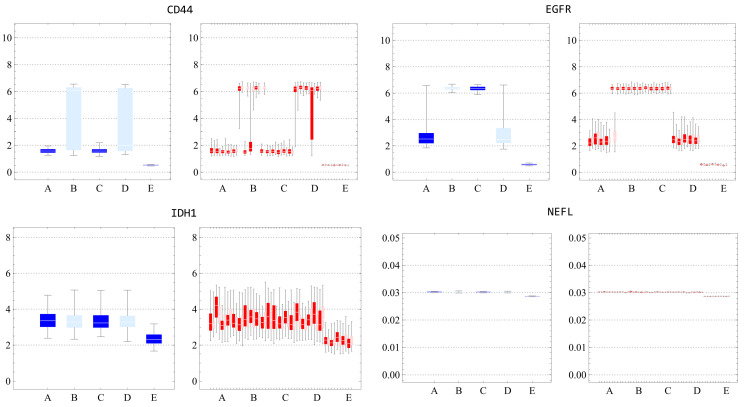
Boxplots for comparison of sample and time averages for all clusters and the four marker genes. The horizontal axis represents the cluster labels, and the vertical axis shows the expression values. The figure displays two sets of plots: the left set shows the sample average of 100 samples at the final time interval, while the right set represents the time average considering 10 trajectories from time 30 to 50 (steps 300 to 500).

**Table 1 ijms-25-04894-t001:** Table with optimized parameters. The first and second lines correspond to parameters after k-means and Neighborhood Contraction clustering, respectively. The left side parameters correspond to the optimization using deterministic dynamics, and the superscript 0 and 1 in $R_{\infty }$ and $R_{1}$ inform if it is for the first or second estimation. The right side parameters correspond to the optimization considering stochastic dynamics, with G/C as the mean gene number per cluster when considering the optimization of the parameters.

*n*	h(x)	$f_{a}$	$f_{b}$	$R_{\infty }^{\;0}$	$R_{\infty }^{\;1}$	$R_{1}^{\;0}$	$R_{1}^{\;1}$	$c_{0}$	*a*	sa	*b*	sb	G/C
1	2	0.1	1.3	1.62	1.15	24.85	20.89	3.5	1.4	0.7	0.9	1.3	20.40
1	2	0.1	1.1	1.72	1.69	35.43	28.48	5.6	1.4	0.6	1.3	1.2	16.71

## Data Availability

The codes developed for the analysis presented in this report are available in the provided GitHub repository. All data utilized to generate the plots are shared in the Appendix A.

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
