# Peer review of "Unveiling the Dynamics behind Glioblastoma Multiforme Single-Cell Data Heterogeneity"

_ijms, 2024, doi:10.3390/ijms25094894_

Round 1
Reviewer 1 Report
Comments and Suggestions for Authors
This manuscript focused on analyzing the single-cell RNA-sequence (scRNA-seq) data to investigate the dynamics of Glioblastoma Multiforme (GBM). The novelty of the proposed bioinformatics method is to utilize the “cluster centroids” as the fixed coordinates for the estimation of parameters, and multiple factors, such as genomic instability were included for characterizing the dynamics through the fixed coordinates.
The manuscript was clear written and the results were supported by a variety of data. One minor issue that the reviewer found is the transition graph in Fig. 6b. The transition weights should be normalized, i.e., the sums of the incoming/ouot-coming probabilities for each node should be 1.
Reviewer 2 Report
Comments and Suggestions for Authors
The work advances the modeling of 15 gene expression dynamics and paves the way for personalized therapeutic interventions by performing several simulations based upon sample and time averages equivalence. The report provides a data-driven quantification of the 196 (epi)genetic landscape specific to GBM and its respective subtypes. The simulation quite complex. The paper contribute to the biological system field by considering factors like genomic instability to characterize the cancer dynamics through stochastic fixed points. The paper can be accepted in the present form.
Reviewer 3 Report
Comments and Suggestions for Authors
The manuscript discusses the aggressive nature of Glioblastoma Multiforme, a type of brain tumor, and proposes a method to interpret the heterogeneity of single-cell RNA-sequence data (scRNA-seq) within the cancer attractors domain. By considering factors such as genomic instability, the study characterizes cancer dynamics through stochastic fixed points derived from centroids obtained via clustering methods. The study also examines stability and transitions between attractors, revealing potential interactions between subtypes related to cancer recurrence and progression. This work bridges molecular mechanisms of cancer heterogeneity with gene expression dynamics, advancing understanding and potentially enabling personalized therapeutic interventions. Although the paper is nicely organized and well written, there was still an issue to be improved. The current single-cell sequencing analysis methods have become relatively mature, so the author needs to point out where the advantages of their method lie compared to previous methods.
